# An analytical model of "Electron-Only" magnetic reconnection rates
Yi-Hsin Liu [1] ✉, Prayash Pyakurel[2], Xiaocan Li[3], Michael Hesse[4], Naoki Bessho[5], Kevin Genestreti[6] & Shiva B. Thapa[1]

"Electron-only" reconnection, which is both uncoupled from the surrounding ions and much faster than standard reconnection, is arguably ubiquitous in turbulence. One critical step to understanding the rate in this novel regime is to model the outflow speed that limits the transport of the magnetic flux, which is super ion Alfvénic but significantly lower than the electron Alfvén speed based on the asymptotic reconnecting field. Here we develop a simple model to determine this limiting speed by taking into account the multiscale nature of reconnection, the Hall-mediated electron outflow speed, and the pressure buildup within the small system. The predicted scalings of rates and various key quantities compare well with fully kinetic simulations and can be useful for interpreting the observations of NASA's Magnetospheric-Multiscale (MMS) mission and other ongoing missions.

Magnetic reconnection converts magnetic energy into plasma thermal and kinetic energy in laboratory, space, and astrophysical plasmas. Recently, NASA's magnetospheric-multiscale (MMS) mission[1] discovered a novel form of reconnection in the turbulent magnetosheath downstream of Earth's bow shock[2–5]. These reconnection events, characterized by electron-scale current sheets with super ion-Alfvénic electron jets and no ion outflows, were named "electron-only" reconnection. The ions are decoupled from the system because of a limited spatial and temporal span dictated by the scale of turbulence eddies[6–9]. Electron-only reconnection has also been identified in other regions, including the bow shock transition layer[10–12] and its foreshock[13], Earth's magnetotail[14–16], macro-scale magnetic flux ropes[17], reconnection exhausts[18], dipolarization fronts[19], and has been studied in laboratory experiments[20–23]. One pronounced feature of such reconnection events, which is not fully understood, is their higher rates in processing magnetic flux and releasing magnetic energy than standard reconnection.

Using particle-in-cell (PIC) simulations, Pyakurel et al.[6] suggested that the transition from standard, ion-coupled reconnection to electron-only reconnection occurs when the system size is smaller than $\sim \mathcal{O}(10)$ ion-inertial ($d_i$) scales, which appears to be consistent with MMS analyses[3,4]. In another independent numerical study, Guan et al.[24] showed that the ion gyro-radius ($\rho_i$) is also critical in controlling this transition.

In light of these PIC simulations, in this work, we model the underlying physics that enables the faster flux transport in the electron-only regime, namely the electron outflow speed. This speed is not limited by the ion Alfvénic speed when ions are not coupled within the system, unlike that in the standard reconnection. The electron outflow speed not only determines the magnetic flux transport into the reconnection exhausts but also the

geometry surrounding the electron diffusion region (EDR), where the magnetic flux frozen-in condition for electron flows is violated[25–27]. To derive this speed, the analytical model presented here incorporates both the dispersive nature of the electron jets within the Hall regime[28–30] and the back pressure accumulated at the outflows. We found that both effects are encoded in the in-plane electric field, which is important to the acceleration of electrons. The resulting scalings of various key quantities in different system sizes compare well with those in PIC simulations. The leading outcome of this theory is the explanation of why the normalized electron-only reconnection rate appears to be bounded by a value $\simeq \mathcal{O}(1)$ in a closed system, as seen in PIC simulations. Besides, it also predicts a higher upper bound value $\simeq 4.28$ if the outflow boundary is open.

## Results

To highlight key features critical to the rate determination, we carry out 2D PIC simulations of magnetic reconnection in plasmas of realistic proton-to-electron mass ratio $m_i/m_e = 1836$. We employ the setup of case A by Pyakurel et al.[6] that has a guide field $B_g = -8B_{x0}$, where $B_{x0}$ is the reconnecting component. The ion $\beta_i = 3.54$ and electron $\beta_e = 0.35$. These are chosen based on the parameters of the MMS electron-only event[2], but with five different system sizes, $L_x \times L_z = 1.28d_i \times 2.56d_i$, $2.56d_i \times 2.56d_i$, $3.84d_i \times 3.84d_i$, $5.12d_i \times 5.12d_i$, and $7.68d_i \times 7.68d_i$. Details of the simulations setup are in the "Methods" section. The units used in the presentation include the ion cyclotron time $\Omega_{ci}^{-1} \equiv (eB_{x0}/m_i c)^{-1}$, the in-plane ion Alfvén speed $V_{Ai0} = B_{x0}/(4\pi n_0 m_i)^{1/2}$ based on the upstream density $n_0$, and the ion inertial length $d_i \equiv c/(4\pi n_0 e^2/m_i)^{1/2}$.

[1]Department of Physics and Astronomy, Dartmouth College, Hanover, NH, USA. [2]Space Sciences Laboratory, University of California, Berkeley, CA, USA. [3]Los Alamos National Laboratory, Los Alamos, NM, USA. [4]Ames Research Center, NASA, Moffett Field, CA, USA. [5]Goddard Space Flight Center, NASA, Greenbelt, MD, USA. [6]Southwest Research Institute, Durham, NH, USA. ✉e-mail: Yi-Hsin.Liu@Dartmouth.edu

## Character of "electron-only" reconnection

PIC simulations capture electron-only reconnection when the domain size is small enough. Figure 1 shows the essential features in the $L_x = 2.56d_i$ case. The electron outflow speed $V_{ex}$ (Fig. 1a) indicates active transport of reconnected magnetic flux. Unlike in ion-coupled standard reconnection, it is evident that ion outflows $V_{ix}$ do not develop in Fig. 1b. Interestingly, electron-only reconnection has a higher reconnection rate than the standard reconnection rate of $\mathcal{O}(0.1)$[31–34], as shown in Fig. 1e. This is somewhat expected because magnetic flux transport is now not limited by the ion Alfvén speed, as in the ion-coupled reconnection, but by the faster electron Alfvén speed since ions are not magnetized/coupled within the small domain. Naively, if the estimate of the typical EDR aspect ratio ~0.1 times the ratio of the electron Alfvén speed $V_{Ae0} = B_{x0}/(4\pi n_0 m_e)^{1/2}$ and the ion Alfvén speed $V_{Ai0}$ is used, we get the normalized reconnection rate

$$R \equiv \frac{cE_R}{B_{x0}V_{Ai0}} \simeq 0.1 \times \frac{V_{Ae0}}{V_{Ai0}} = 0.1 \times \sqrt{1836} \simeq 4.28 \quad (1)$$

where $E_R$ is the reconnection electric field. Note that, throughout this paper, the subscript "0" is reserved for upstream asymptotic values. This $R$ value, however, is too high compared to the simulation results, as shown in Fig. 1e. The rate only gets closer to unity $\mathcal{O}(1)$, and a scaling law has not been developed yet.

To address this issue, one key observation is that the limiting speed is actually much lower than the asymptotic electron Alfvén speed $V_{Ae0}$. Figure 1c shows cuts of the $x$-direction electron flow velocity $V_{ex}$ in blue, ion flow velocity $V_{ix}$ in red, and the $E \times B$ drift velocity in black along the midplane ($z = 0$). Electrons reach a peak outflow speed $V_{ex,peak} \simeq 0.15V_{Ae0}$ when they exit the EDR (the red box in Fig. 1b). This $V_{ex,peak}$ value (also shown as the purple horizontal line in Fig. 1d) is, instead, close to the electron Alfvén speed based on the local $B_x$ at the EDR-scale in the nonlinear stage; this can be seen by comparing it with the blue line in Fig. 1d near the edge of the red shaded vertical band of $d_e$-scale. We will denote this relation by $V_{ex,peak} \sim V_{Ae} \equiv B_{xe}/(4\pi n m_e)^{1/2}$.

Farther downstream in Fig. 1c, $V_{ex}$ plateaus to a super ion Alfvénic value of $1.7V_{Ai0}$ that is only 4% of the asymptotic electron Alfvén speed $V_{Ae0}$. This critical speed limits the flux transport. The time evolution of the electron outflow velocity $V_{ex}$ cuts (Fig. 2b), demonstrates the development of the plateauing of $V_{ex}$ after the reconnection rate also reaches its plateau (Fig. 1e). Similar $V_{ex}$ plateaus (of different values) also develop in other four simulations of different system sizes, as shown in rest panels of Fig. 2. Note that the plateau in the smallest system ($L_x = 1.28d_i$) in Fig. 2a is less clear due to the back-pressure that will be discussed later. Overall, it is expected that a lower flux transport speed leads to a reconnection rate lower than the estimation in Eq. (1). We will denote this limiting speed as $V_{out,e}|_{L_0}$, which is, the electron outflow speed at a distance $L_0$ downstream of the X-line. Farther downstream of this location, the exhaust opening angle quickly decreases to 0, as marked in Fig. 1b.

## The limiting speed of the flux transport

The first goal is to derive this limiting speed $V_{out,e}|_{L_0}$. We start from the electron momentum equation in the steady state

$$nm_e\mathbf{V}_e \cdot \nabla\mathbf{V}_e = \frac{\mathbf{B} \cdot \nabla\mathbf{B}}{4\pi} - \frac{\nabla B^2}{8\pi} - en\mathbf{E} - \nabla \cdot \mathbb{P}_e. \quad (2)$$

The term on the left-hand side (LHS) is the electron flow inertia. The terms on the right-hand side (RHS) are the magnetic tension force, magnetic pressure gradient force, electric force, and the divergence of the electron pressure, respectively. Note that the ion flow velocity $|\mathbf{V}_i| \ll$ electron velocity $|\mathbf{V}_e|$ condition (i.e., ions do not carry the electric current $\mathbf{J}$) and Ampère's law were used to turn the Lorentz force—$e\mathbf{V}_e \times \mathbf{B}/c \simeq \mathbf{J} \times \mathbf{B}/(nc)$ into the two magnetic forces in Eq. (2). Balancing the electron flow inertia with the

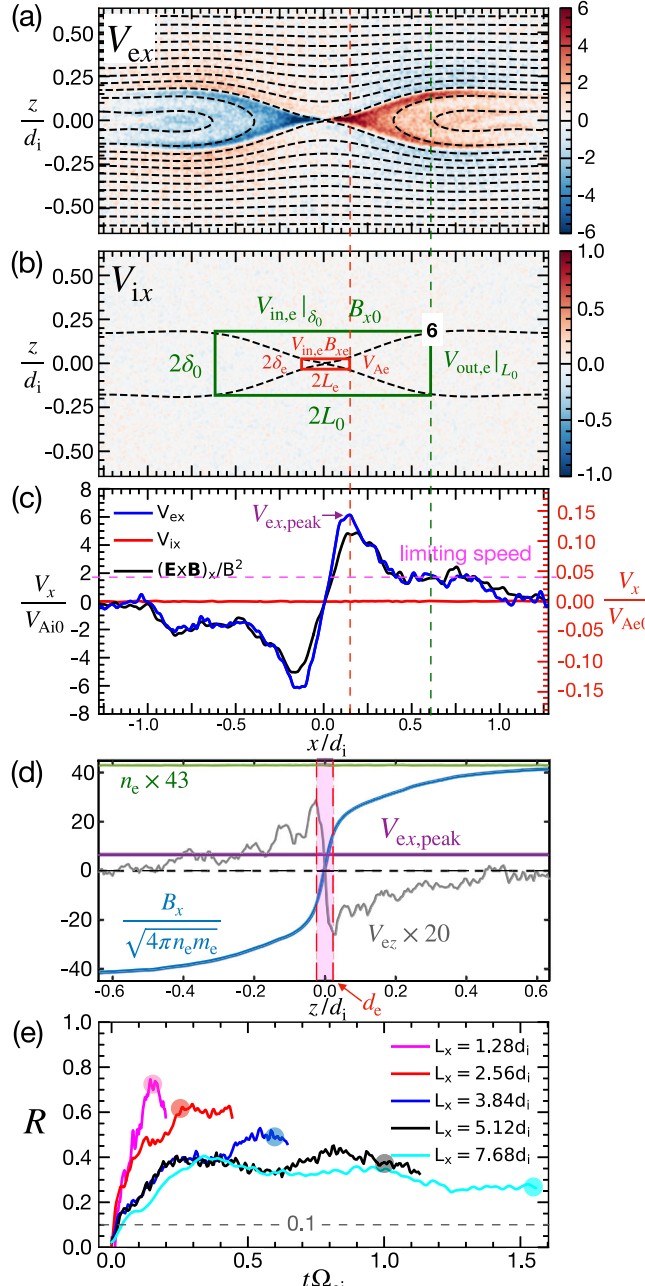

**Fig. 1 | Key features in the $L_x = 2.5d_i$ case and reconnection rates. a** Electron outflow speed $V_{ex}$ overlaid with the contour of the in-plane magnetic flux $\psi$. Note that the entire domain is smaller than the typical ion diffusion region (IDR) in standard reconnection. **b** Ion outflow speed $V_{ix}$ overlaid with the separatrices in dashed black. The red box of size $2L_e \times 2\delta_e$ marks the electron diffusion region (EDR). The corners (such as point "6") of the green box of size $2L_0 \times 2\delta_0$ mark the locations downstream of which the exhaust opening angle quickly decreases to 0. **c** Cuts of $V_{ex}$, $V_{ix}$ and the $E \times B$ drift speed along the $z = 0$ line. The (red and green) dashed vertical lines mark the outflow boundaries of the EDR and the green box in (**b**), while the magenta dashed horizontal line denotes the limiting speed. **d** In blue the electron Alfvén speed based on the local $B_x$ and $n_e$ as a function of $z$ at $x = 0$. In gray the electron inflow speed $V_{ez} \times 20$. In green the electron density $n_e \times 43$. In purple the peak velocity $V_{ex,peak}$ from (**c**). The red shaded band marks the EDR. **e** Reconnection rate $R$ as a function of time for simulations of different system sizes. The rates in our simulations are computed from $R = (\partial\Delta\psi/\partial t)/B_{x0}V_{Ai0}$ where $\Delta\psi$ is the magnetic flux difference between the X-line and the O-line. Note that $\partial\Delta\psi/\partial t = cE_R$, the reconnection electric field, in 2D systems. The gray dashed horizontal line indicates the typical rate of ion-coupled standard reconnection[31]. The transparent color circles mark the time of these $V_{ex}$ contours in Fig. 2.

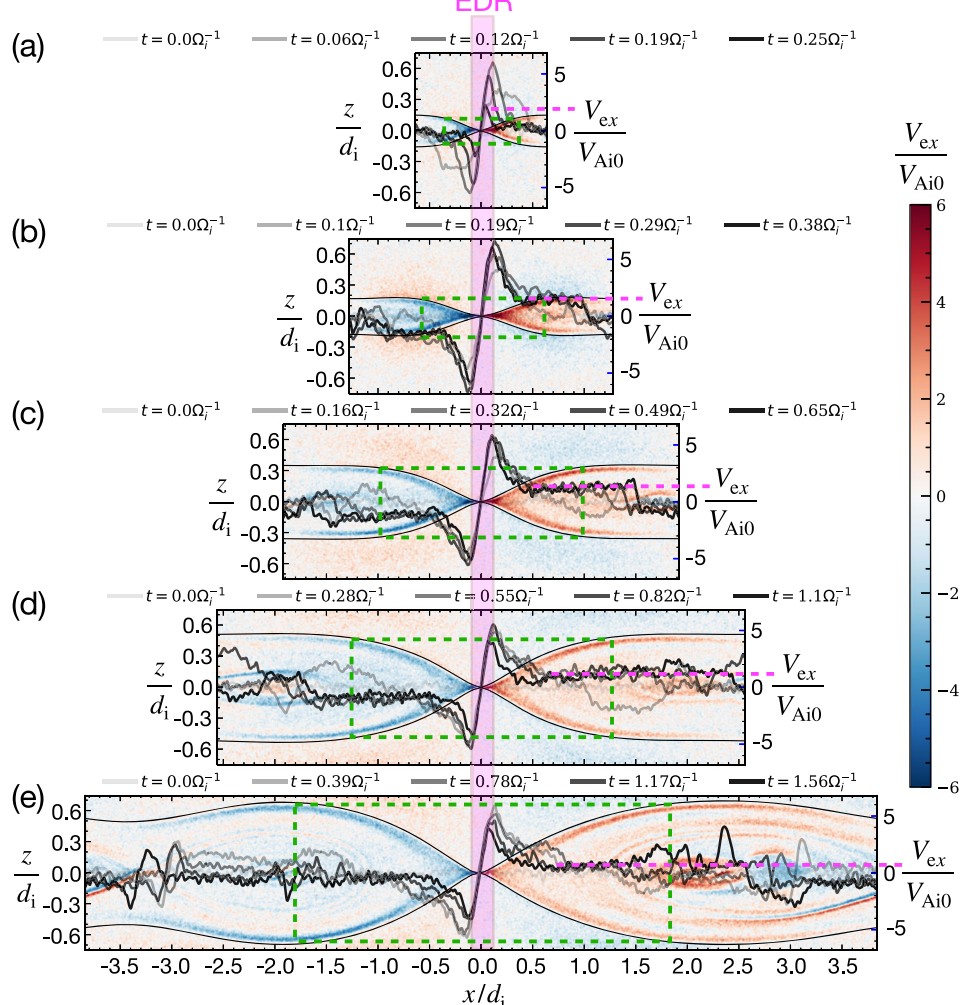

**Fig. 2 | Limiting speed of the flux transport.** The time evolution of $V_{ex}$ cuts at $z = 0$ overlaid on top of $V_{ex}$ contour in simulations of box sizes **a** $L_x = 1.28d_i$ **b** $L_x = 2.56d_i$ **c** $L_x = 3.84d_i$ **d** $L_x = 5.12d_i$ **e** $L_x = 7.68d_i$. The value of these $V_{ex}$ curves can be read by the axis at the right boundary of each panel and the magenta dashed horizontal line shows the representative plateau speed. The time of these $V_{ex}$ cuts is shown on top of each panel while the time of the $V_{ex}$ contour is marked by the corresponding transparent color circle in Fig. 1e. The separatrices are marked in solid black. The red-shaded band marks the electron diffusion region (EDR). The corners of the green boxes denote the locations downstream of which the exhaust opening angle quickly decreases to 0.

magnetic tension $\mathbf{B} \cdot \nabla \mathbf{B}/4\pi$ will lead to an electron jet moving at the electron Alfvén speed. However, the jet can be slowed down by other terms on the RHS, especially the in-plane electric field $\mathbf{E}$. One important source is the Hall electric field $\mathbf{E}_{Hall} = \mathbf{J} \times \mathbf{B}/enc$ that arises from the separation of the lighter electron flows from the much heavier ion flows. $\mathbf{E}_{Hall}$ acts to slow down electrons and speed up ions to self-regulate itself[35]; thus, we expect $E_x$ pointing in the same direction as the outflows that slow down the electron jet[36,37].

To quantify this phenomenon, we take the "finite-difference approximation" of Eq. (2) at point "1" in Fig. 3a. In the $x$-direction, the momentum equation reads

$$\frac{nm_e V_{ex3}^2}{2L_0} \simeq \frac{B_{z1}}{4\pi\delta_0} 2B_{x7} - \frac{B_{z3}^2}{8\pi L_0} - enE_{x1}, \tag{3}$$

where the targeted quantity $V_{ex3}$ is $V_{ex}$ at point "3", etc. Being similar to the analysis from Fig. 1c of Liu et al.[38], this equation, moreover, includes the in-plane electric field critical to the acceleration of electron outflows within the Hall region. This approach allows one to derive the algebraic relation between key quantities while considering the magnetic geometry of the

system[35,38–40]. Here, we ignored the electron pressure gradient and the $B_y^2$ gradient along path 2–3. These are justified since $\Delta P_{exx}$ and $\Delta(B_y^2)/8\pi$ are relatively small[37,41] compared to $B_{x0}^2/8\pi$ ($\propto$ tension) in Fig. 3b.

To estimate $E_{x1}$, we analyze the steady-state Faraday's law $\oint \mathbf{E} \cdot d\vec{\ell} = 0$ and the original momentum equation along the closed loop (2-3-4-5-2) in Fig. 3a. Unlike path 2–3, the flow inertia $|nm_e \mathbf{V}_e \cdot \nabla \mathbf{V}_e|$ along the integral path 3-4-5-2 is negligible compared to $|\mathbf{B} \cdot \nabla \mathbf{B}/4\pi - \nabla B^2/8\pi| = |\mathbf{J} \times \mathbf{B}/c| \simeq |en\mathbf{V}_e \times \mathbf{B}/c|$, so we can write

$$c\int_2^3 E_x dx = -c\int_{3452} \mathbf{E} \cdot d\vec{\ell} \simeq \int_3^4 (\underbrace{V_{ex}B_y - V_{ey}B_x}_{\text{(a)}})dz$$

$$+ \int_4^5 (\underbrace{V_{ey}B_z - V_{ez}B_y}_{\text{(b)}})dx + \int_5^2 (\underbrace{V_{ex}B_y}_{\text{(c)}} - \underbrace{V_{ey}B_x}_{\text{(d)}})dz. \tag{4}$$

Term (b) vanishes since $V_{ey} = 0$ at the upstream; term (c) vanishes because $V_{ex} = 0$ along the inflow symmetry line. Terms (a) and (d) roughly cancel each other because $\int V_{ey}B_x dz \propto \int J_y B_x dz \propto \int (\partial_z B_x)B_x dz = \Delta(B_x^2)/2$, which is $B_{x0}^2/2$ for the 3–4 and $-B_{x0}^2/2$ for the 5–2 integral paths.

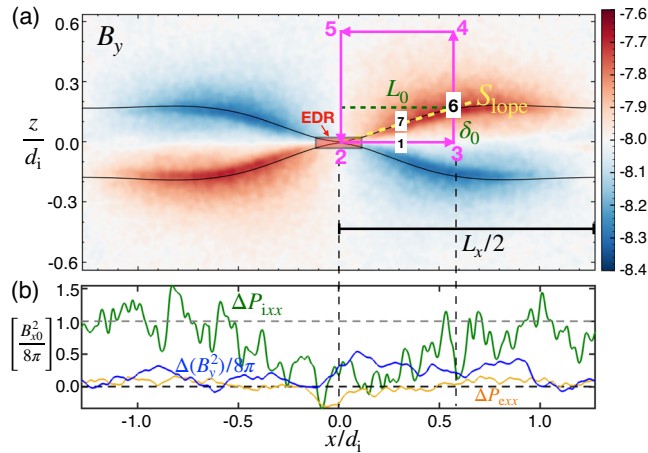

Fig. 3 | Quantities critical to the estimation of the in-plane electric field. a The out-of-plane magnetic field $B_y$ (i.e., showing the Hall quadrupole signature) and the integral path of Eq. (4) in magenta. The red-shaded region marks the electron diffusion region (EDR), and the black solid curves trace the magnetic separatrices. Critical points and the separatrix slope ($S_{slope} = \delta_0/L_0$) used in the analysis are annotated. b The difference of pressures from their upstream asymptotic values for components $\Delta P_{ixx}$ (in green), $\Delta P_{exx}$ (in yellow) and $\Delta(B_y^2)/8\pi$ (in blue) along the $z = 0$ line. For reference, $B_{x0}^2/8\pi$ is plotted as the gray dashed horizontal line. While the oscillation in the $\Delta P_{ixx}$ curve is unavoidable because of the noise in hot ions, the pressure depletion at the X-line is discernible.

This equation can then be approximated as

$$c\frac{E_{x1}}{2}L_0 \simeq V_{ex3}\int_3^6 B_y\,dz - B_{y0}\int_4^5 V_{ez}\,dx. \quad (5)$$

The LHS used the fact that $E_x$ increases monotonically from 0 at the X-line to point "3." The first integral on the RHS holds because the outflow $V_{ex}$ is narrowly confined within the separatrices. In the next step, we further approximate $\int_3^6 B_y\,dz \simeq [(B_{y6} + B_{y3})/2]\delta_0$. And, the last integral $\int_4^5 V_{ez}\,dx \simeq \int_3^4 V_{ex}\,dz \simeq V_{ex3}\delta_0$, since the particle fluxes going through sides 2–3 and 2–5 are negligible due to the symmetry shown in Fig. 3a and incompressibility is used. With the upstream $B_{y0} \simeq B_{y3}$ as in Fig. 3a, we can then combine the two terms on the RHS to derive

$$E_{x1} \simeq \frac{V_{ex3}}{c}(B_{y6} - B_{y3})\frac{\delta_0}{L_0} \simeq \frac{4\pi ne}{c^2}\frac{\delta_0^2}{L_0}V_{ex3}^2. \quad (6)$$

Here the last equality used Ampère's law $(B_{y6} - B_{y3})/\delta_0 \simeq (4\pi/c)neV_{ex3}$. We note that the electric field $E_{x1}$ is basically determined by the convection of the Hall magnetic quadrupole field (i.e., $B_{y6} - B_{y3}$) and $\int_{23452}\mathbf{E}\cdot d\vec{\ell} = 0$, as illustrated in Fig. 4a.

While this model mimics the characteristics of the electron current system of an idealized exhaust, it does not consider the effect of the closed boundary, which can be significant in a small system. In particular, the high ion pressure originating from the initial current sheet will accumulate into the plasmoid at a fixed location. With nearly immobile ions, where $nm_i\mathbf{V}_i\cdot\nabla\mathbf{V}_i$ is negligible compared to other forces in the ion momentum equation, $en\mathbf{E} \simeq \nabla P_i$[37,41], as illustrated in Fig. 4b. In the small system size limit, one would expect that $enE_{x1} \simeq (P_{i3} - P_{i2})/L_0$ can be easily of the order of $B_{x0}^2/(8\pi L_0)$ due to the build-up of pressure within the plasmoid and the depletion of the pressure component $xx$ at the X-line[35], as shown by the central dip in the $\Delta P_{ixx}$ (green) curve of Fig. 3b.

Hence, we will impose a reasonable condition where the sum of the plasma and magnetic pressures completely cancels the magnetic tension in the $L_x \to 0$ limit. This can be done by including this ion back pressure into

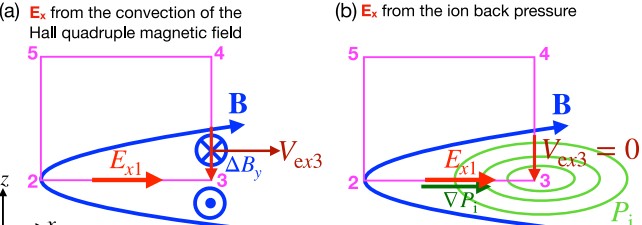

Fig. 4 | Sources of the in-plane electric field $E_{x1}$. a The motional electric field $-V_{ex3}\Delta B_y/c$ arising from the convection of the Hall magnetic quadrupole field $\Delta B_y \equiv B_y - B_g$, combined with the steady-state Faraday's law $\int_{23452}\mathbf{E}\cdot d\vec{\ell} = 0$; this corresponds to the $f \to 0$ limit discussed in Eq. (7). b The ion back pressure accumulated within the plasmoid. Here the $P_i$ contour is illustrated in green; this corresponds to the $f \to 1$ limit discussed in in Eq. (7).

the full $E_{x1}$ using a function $f(L_x)$,

$$E_{x1} \simeq \frac{4\pi ne}{c^2}\frac{\delta_0^2}{L_0}V_{ex3}^2 + f\frac{B_{x0}^2 - B_{z3}^2}{8\pi L_0 ne}. \quad (7)$$

We choose $f(L_x) = \mathrm{sech}(L_x/\Delta_f)$ so that, for $L_x \gg \Delta_f$ then $f \to 0$, corresponding to Fig. 4a. For $L_x \ll \Delta_f$ then $f \to 1$, where the outflow is shut off and the ion pressure gradient dominates, as in Fig. 4b. The length scale $\Delta_f$ will later be determined to be $\Delta_f = 1.28d_i$, and the $f$-profile is shown in Fig. 5b. The ion–electron interaction is primarily mediated by the electric field within the Hall region. Hence, it seems appropriate to heuristically include the effect of ion back pressure into the electric field estimation.

Plugging Eq. (7) back to Eq. (3), and realizing $B_{z1} \simeq B_{z7} \simeq (\delta_0/L_0)B_{x7}$, the separatrix slope $S_{slope} \simeq \delta_0/L_0$, $B_{z3} \simeq 2B_{z1}$, $B_{x7} \simeq B_{x6}/2$, and $B_{x6} \simeq B_{x0}$ from the magnetic field line geometry (see the flux function contour in Fig. 1a), we obtain the limiting speed

$$V_{out,e}|_{L_0} = V_{ex3} \simeq \frac{d_i}{\delta_0}V_{Ai0}\sqrt{\frac{(1 - S_{slope}^2)(1 - f)}{2 + (d_e/\delta_0)^2}}. \quad (8)$$

A critical feature in Eq. (8) is $V_{out,e}|_{L_0} \propto \delta_0^{-1}$, which provides a faster jet in a narrower exhaust. Without the corrections gathered within the square root, if $\delta_0 \to d_e$ then $V_{out,e}|_{L_0} \to V_{Ae0}$ (i.e., also true for $\delta_0 \ll d_e$ when the electron inertial effect $(d_e/\delta_0)^2$ within the square root is retained). This is responsible for the faster flux transport speed at sub-$d_i$-scales, but it transitions to the ion Alfvén speed when $\delta_0 \to d_i$, because $V_{out,e}|_{L_0} \to V_{Ai0}$, as in ion-coupled standard reconnection. In the limit $\delta_0 \gg d_i$, one needs to consider the full two-fluid equations [e.g. ref. 42], coupling ions back to the scale larger than the typical ion diffusion region (IDR) size. The resulting $V_{out,e}|_{L_0}$ remains ion Alfvénic [e.g. ref. 43].

This scale-dependent velocity is the dispersive property discussed in the idea of Whistler/Kinetic Alfvén wave (KAW)-mediated reconnection[28–30,42,44], but here we also include the reduction by the back pressure (parameterized by $f$) within a small system. The flow is stopped when $f \to 1$ in Eq. (8), corresponding to the limit $L_x \ll \Delta_f$ where the total pressure gradient completely cancels the tension force in Eq. (3). Finally, the outflow speed is also reduced with a larger opening angle ($S_{slope}\uparrow$).

## Geometry and reconnection rates

This limiting speed not only determines how fast magnetic flux is convected into the outflow exhaust but also the upstream magnetic geometry and, thus, the strength of the reconnecting magnetic field immediately upstream of the EDR. All together, one can derive the electron-only reconnection rate.

We closely follow the approach in ref. 35 to estimate the magnetic field strength $B_{xe}$ immediately upstream of the EDR of size $2L_e \times 2\delta_e$, as marked

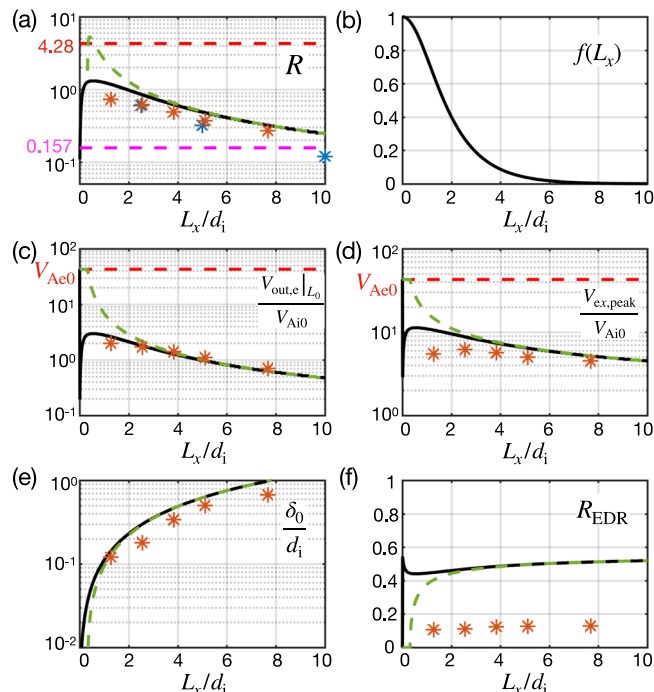

**Fig. 5 | Predictions as a function of the system size $L_x$. a** The normalized reconnection rate. **b** The profile $f(L_x) = \text{sech}(L_x/1.28d_i)$ used for the black solid curves in other panels. **c** The limiting speed of flux transport. **d** The peak electron outflow speed. **e** The exhaust half-thickness. **f** The rate normalized to the electron diffusion region (EDR) quantities. The predictions with $f(L_x)$ in (**b**) are shown as the black solid curves, while the green dashed curves have $f = 0$. Orange symbols are from the PIC simulations carried out in this paper. In (**a**), the blue symbols are from ref. 6. For comparison, the rough prediction from Eq. (1) is marked by the red dashed horizontal line, and $R = 0.157$ predicted for ion-coupled standard reconnection[35] as the magenta dashed horizontal line. In (**c**) and (**d**), the maximum plausible electron outflow value, $V_{Ae0}$, is marked as red horizontal dashed lines.

by the red box in Fig. 1b and $\delta_e \sim d_e$. One can write

$$\frac{cE_{ye}}{B_{xe}V_{Ae}} = \frac{V_{in,e}}{V_{Ae}} \simeq \frac{\delta_e}{L_e} \sim \frac{\delta_0}{L_0} \simeq \frac{V_{in,e}|_{\delta_0}}{V_{out,e}|_{L_0}} = \frac{cE_y|_{\delta_0}}{B_{x0}V_{out,e}|_{L_0}}, \quad (9)$$

where $L_0$ and $\delta_0$ are the exhaust length and half-width. Other relevant quantities are annotated in Fig. 1b. For instance, $V_{in,e}$ is the electron inflow speed at $z = \delta_e$ while $V_{in,e}|_{\delta_0}$ is the value at $z = \delta_0$. The first equality of Eq. (9) used the frozen-in condition upstream of the EDR. The second equality holds because of the incompressibility and $V_{ex,peak} \simeq V_{Ae}$. The third equality approximates the separatrix as a straight line to simplify the geometry. The fourth and fifth equalities used similar arguments to the quantities at the edge of the larger $L_0 \times \delta_0$ box. Finally, in the 2D steady-state, $E_y$ is uniform. Thus, the equality between the first and the last terms gives,

$$V_{out,e}|_{L_0} \simeq \frac{B_{xe}}{B_{x0}} V_{Ae} = \left(\frac{B_{xe}}{B_{x0}}\right)^2 \left(\frac{m_i}{m_e}\right)^{1/2} V_{Ai0}. \quad (10)$$

An important difference from Liu et al.[35] is that $B_{xi}$ in their Eq. (5) is now replaced by $B_{x0}$, since the entire system is within the IDR.

Liu et al.[35] further estimated the depletion of the pressure component along the inflow direction, caused by the vanishing energy conversion $\mathbf{J} \cdot \mathbf{E}_{Hall} = \mathbf{J} \cdot (\mathbf{J} \times \mathbf{B}/nec) = 0$; note that $\mathbf{E}_{Hall}$ dominates within the IDR and this pressure depletion provides the localization mechanism necessary for

fast reconnection. One can then use force balance along the inflow direction to relate $B_{xe}$ to the separatrix slope $S_{slope}$[35]. In the case where the guide field at the X-line does not change much from its upstream value, like $B_{y2}$ in Fig. 3, we get

$$\frac{B_{xe}}{B_{x0}} \simeq \frac{1 - 3S_{slope}^2}{1 + 3S_{slope}^2}. \quad (11)$$

The only difference is again that $B_{xi}$ in Eq. (9) of Liu et al.[35] is now replaced by $B_{x0}$. In order to get the full solution from Eqs. (8), (10), and (11), one still needs to relate $\delta_0$ to $S_{slope}$. We approximate

$$\delta_0 = L_0 S_{slope} \sim 0.5 \left(\frac{L_x}{2}\right) S_{slope}, \quad (12)$$

as it is reasonable to expect $2L_0$ to be on the order of the system size $L_x$, as in Fig. 1b. We can then equate Eqs. (8) and (10) and solve for $S_{slope}$ numerically.

Once $S_{slope}$ is determined, we can estimate the normalized reconnection rate,

$$R \equiv \frac{cE_R}{B_{x0}V_{Ai0}} \simeq \frac{V_{out,e}|_{L_0} B_{z3}}{B_{x0}V_{Ai0}} \simeq \frac{V_{out,e}|_{L_0}}{V_{Ai0}} S_{slope}. \quad (13)$$

The last equality used $B_{z3}/B_{x0} \simeq B_{z6}/B_{x6} \simeq S_{slope}$. In Fig. 5a, the prediction of $R$ as a function of $L_x$ without including the back pressure effect (i.e., $f = 0$) is shown as the green dashed curve, while the prediction with nonzero $f(L_x)$ (given in Fig. 5b) is shown as the black solid curve. In a similar format, the limiting speed (Eq. (8)) is shown in Fig. 5c, while the more pronounced peak electron jet speed $V_{ex,peak} \simeq V_{Ae} = (B_{xe}/B_{x0})(m_i/m_e)^{1/2}V_{Ai0}$ is shown in Fig. 5d. The estimated exhaust width (Eq. (12)) is shown in Fig. 5e. Simulation results are plotted as orange symbols, whose values can be read off from Figs. 1e and 2.

Overall, the green dashed curves already work reasonably well for $2.56d_i \lesssim L_x \lesssim 10d_i$ cases, but they overestimate quantities in the $L_x = 1.28d_i$ case. For this reason, we set the length scale $\Delta_f = 1.28d_i$ in $f(L_x)$ to parametrize the back pressure effect that suppresses the outflow and rate. This corrects the predictions, and the resulting black solid curves capture the scaling of these key quantities in Fig. 5a, c, d, e; the quantitative agreements are within a factor of 2. Importantly, the rate ($R$) is now bounded by a value $\simeq \mathcal{O}(1)$, addressing the key question that motivates this work.

## Discussion

A framework for predicting the electron-only reconnection rate (Eqs. (8), (10), (11), (12), and (13)) is developed after recognizing the difference in the EDR-scale and the asymptotic regions, considering both the inflow and outflow force-balances within the ion inertial scale. This simple model not only provides reasonable predictions for the simulated rates in kinetic plasmas but also captures the scaling of various key quantities in PIC simulations of different sizes (Fig. 5). We find that the in-plane electric field (Fig. 4) regulates the electron outflow speed and thus the reconnection rates. It is worth mentioning that this model has successfully integrated the idea of Whistler/KAW-mediated reconnection[28–30,42,44] into the reconnection rate model[35].

For in-situ MMS observations, it might be challenging to determine the far upstream, asymptotic magnetic field $B_{x0}$ using the short-scaled tetrahedron formation. Practically, it is more accessible to obtain the rate normalized by the local quantities around the EDR, $R_{EDR} \equiv cE_R/(B_{xe}V_{Ae}) \simeq (B_{xe}/B_{x0})^{-2}(m_i/m_e)^{-1/2}R$. Our theory in Fig. 5f predicts a nearly constant $R_{EDR} \sim 0.4$–$0.5$. In Fig. 1d, one $d_e$ upstream of the X-line is close to the location of the peak electron inflow speed and features the upstream edge of the EDR that MMS can easily identify[34,45,46]. The resulting $R_{EDR}$ (orange symbols in Fig. 5f) based on the measured $B_{xe}$ at $z = 1d_e$ are four times lower (i.e., $R_{EDR} \simeq 0.1$)[47]. However, we also note that the $B_x$ at the location where $V_{ex,peak} = V_{Ae}$

holds accurately is roughly twice smaller than $B_{xe}$ because of the sharp $B_x$ profile at $d_e$-scales (i.e., note that this profile is proportional to the $B_x/\sqrt{4\pi n_e m_e}$ profile in Fig. 1d because of the constancy of $n_e$). If we take this $B_x$ as $B_{xe}$, the factor-of-two difference results in a four-times higher $R_{EDR}$, which may explain this discrepancy. Despite this extra complexity, our simple theory captures the constancy of the simulated $R_{EDR}$. Recent MMS observational reports of electron-only reconnection indicate rates around 0.25[45,46]. Another event at the magnetopause suggests an even higher reconnection rate, up to ~0.4 during the onset phase[48].

Even with a strong guide field ($|B_g| = 8B_{x0}$) in our simulation, the ion gyro-radius $\rho_i = 1.23 d_i$ due to the high ion temperature ($T_{i0} = 115.16 m_i V_{Ai0}^2$). Guan et al.[24] studied cases of guide fields $B_g = 1B_{x0}$ and $8B_{x0}$, and they concluded that the $|V_i| \ll |V_e|$ condition is met when the system size is smaller than the ion gyroradius ($\rho_i$). Presumably, because with a high ion thermal speed ($10.73 V_{Ai0}$ in our runs) and large gyro-radii, ions will be quickly gyrated out of the region of constant $\mathbf{E}$, avoiding the formation of coherent ion flows through direct acceleration over a longer time span[47]. Our analytical theory is built on this $|V_i| \ll |V_e|$ condition (i.e., ions do not carry currents as in the EMHD limit[49–51]), and it explains the transition to the standard reconnection rate at $L_x \gtrsim 10 d_i$, as shown by Pyakurel et al.[6]. Under this same condition, the analytical approach (and thus the predictions) derived here also works for anti-parallel reconnection and is not limited to the strong guide field case.

Caveats should be kept in mind when applying these predictions. Related to the above discussion, our theory does not model the lower rate reported with a small ion gyro-radius $\rho_i (\ll L_x)$ where ion currents emerge, as reported in Guan et al.[24]. Bessho et al.[12] found $cE_R/(B_{x0}V_{ex,peak})$ ranging from 0.1 to 0.7 in the turbulent shock transition region, indicating the possibility of a much higher rate, potentially due to the driving of high-speed background flows. In addition, with a non-periodic, open outflow system, such as the merger between isolated small-scale magnetic islands, electron-only reconnection therein may not saturate early due to the back pressure and may achieve a higher rate ($R \simeq 4.28$) as predicted by the green dashed curves in Fig. 5a. Finally, the thickness-dependent growth rate of the tearing instability in this regime may also contribute to its onset and the early development of electron-only reconnection[47,52–55]. Together with the time dependence and the full 3D nature[56], future endeavors are required to develop a more complete theory. Nevertheless, our simple model demonstrates a working framework addressing critical features that necessitate faster electron-only reconnection rates.

## Methods

We carry out 2D PIC simulations of magnetic reconnection in proton-electron plasmas with mass ratio $m_i/m_e = 1836$ using the P3D code[57]. We employ the setup of case A by Pyakurel et al.[6], which is designed based on parameters of the MMS electron-only event[2], but with five different system sizes. The double Harris sheet profile $\mathbf{B} = B_{x0}[\tanh(z - 0.25L_z/w_0) - \tanh(z - 0.75L_z/w_0) - 1]\hat{x} + B_g\hat{y}$ is employed, with a uniform guide field $B_g = -8.0B_{x0}$. The initial half thickness $w_0 = 0.06 d_i$ where the ion inertial scale $d_i \equiv (m_i c^2/4\pi n_0 e^2)^{1/2}$ is normalized to the upstream density $n_0$. The in-plane ion Alfvén speed $V_{Ai0} = B_{x0}/(4\pi n_0 m_i)^{1/2}$ and cyclotron frequency $\Omega_{ci} \equiv eB_{x0}/m_i c$ are normalized to the reconnecting component $B_{x0}$. The speed of light $c = 300 V_{Ai0}$. The high temperature $T_{i0} = 115.16 m_i V_{Ai0}^2$ and $T_{e0} = 11.51 m_i V_{Ai0}^2$ result in $\beta_i = 8\pi n_0 T_{i0}/(B_{x0}^2 + B_g^2) = 3.54$ and $\beta_e = 0.35$, and a nearly uniform density from pressure balance condition. The ratio of gyro-radius (based on the full field strength) and inertial length are $\rho_i/d_i \simeq 1.33$ for ions and $\rho_e/d_e \simeq 0.42$ for electrons. This $T_i \gg T_e$ limit is favorable to the occurrence of electron-only reconnection (see the "Discussion" section). The simulation sizes are $L_x \times L_z = 1.28 d_i \times 2.56 d_i$, $2.56 d_i \times 2.56 d_i$, $3.84 d_i \times 3.84 d_i$, $5.12 d_i \times 5.12 d_i$, and $7.68 d_i \times 7.68 d_i$, with cell size $0.21 d_e$ and time step $2.5 \times 10^{-5}\Omega_{ci}^{-1}$. The particle number per cell is 6000. Periodic boundaries are used. In our figures, we show the top current sheet with our coordinate origin re-centered at the X-line.

## Data availability

Access to the simulation data and scripts used to plot the figures is available at Zenodo (https://doi.org/10.5281/zenodo.14919784). All other data are available from the corresponding author upon reasonable request.

## Code availability

The P3D code is available through collaboration with the second author, P.S.P. Upon request, the code's developer grants access to and helps run the simulations and handle the output data. The simulation data are analyzed using IDL and Python. The scripts to read the output data are available at the data storage site Zenodo.org.

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

## Acknowledgements

Y.L. is grateful for support from NASA's MMS mission 80NSSC21K2048 and NSF Career Award 2142430. P.P. is supported by NASA MMS early career grant 80NSSC21K1481 and NASA grant 80NSSC24K0559. We would like to acknowledge the high-performance computing support from Derecho, which is provided by NCAR's Computational and Information Systems Laboratory, sponsored by the National Science Foundation (NSF), and NERSC Advanced Supercomputing, sponsored by the Department of Energy (DOE).

## Author contributions

Y.L. derived the theory and wrote the paper. P.P. carried out the simulation and analysis shown in this paper. X.L. and S.T. carried out additional simulations confirming the general conclusion. M.H. and N.B. provided theoretical inputs. K.G. provides observational inputs. All authors discussed and commented on the paper.

## Competing interests

The authors declare no competing interests.
