## [Transparent Peer Review file · Communications Physics]

An Analytical Model of "Electron-Only" Magnetic Reconnection Rates

Corresponding Author: Professor Yi-Hsin Liu

Version 0:

Version 0:

Reviewer comments:

Reviewer #1

(Remarks to the Author)

The manuscript is well written and the topic very interesting.

It has two weak points that, in my opinion, do not allow the publication on a broad-public journal as Physics Communications:

1) Even if the subject is "broad public", since how fast magnetic reconnection proceed determines the dynamical temporal scale of a variety of magnetized system, the way the analytical estimations for the reconnection rate are presented is too specialistic. It is very hard for a reader, I would not say "not used to magnetic reconnection", but simply "not aware of the approach adopted in Ref. 28, 31-33" (cited by the Authors for justifying their approach) to understand the choice of different indexes for V_{ex} , B_x , etc.

I understand that the number of line is limited for a brief communication, but clearly state the geometry, method, approximations that lead to equation (2), as it has been done in the original articles using a similar technique, is mandatory. Otherwise, the article should be published on far more specialized journals, as JGR or PoP. Part of the Discussion, in particular the lines regarding Fig. 4e, 4d (that provide simple estimations, without numerical comparisons), as well as these two panels, could be sacrificed, at the advantage of a clear and pedagogical explanation of how getting Eq. (2).

2) I disagree with (part) of the estimation leading to Eq. (4).

The d) term in Eq. (3) is comparable with the main terms. I agree on the fact that far from the EDR its contribution is small (and roughly cancels the a) term. On the contrary, its contribution close to/in the EDR is far from negligible. Indeed, even if B_x is small there, V_{ey} is large there. Their product roughly reads

$$-J_y/(n * e) * B_x \sim -(B_{xe} * c)/(4\pi * \delta_e) * 1/(n * e) * B_{xe}$$

where, as in the manuscript δ_e is the EDR width and B_{xe} is the value of B_x at the EDR scale.

Its integral contribution to Eq. (3) reads

$$-\int_0^{\delta_e} V_{ey} B_x dz \sim \int_0^{\delta_e} c * B_{xe}^2 / (4\pi * n * e * \delta_e) dz \sim c * B_{xe}^2 / (4\pi * n * e) \sim c * V_{Ae}^2 * (m_e / e)$$

neglecting a factor of order one, coming from the actual profile of J_y and B_x .

Here $\int_0^{\delta_e} dz \sim \delta_e$ since $V_{ey} B_x$ is small on the rest of the path, and V_{Ae} is, as in the manuscript, the value of V_{Ae} at the EDR scale.

Let's call E_d the contribution of the d) term to E_{x1} . It reads

$$E_d \sim c * V_{Ae}^2 * (m_e / e) * 2/(c * L_0)$$

Hereafter, I'll call E_5 the expression for E_{x1} provided by Eq (5).

The ratio between E_d and E_5 reads

$$E_d / E_5 \sim ((V_{Ae}^2 * m_e) / (L_0 * e)) / ((4\pi * n * e * \delta_e^2 * V_{ex}^2) / (c^2 * L_0)) \sim (V_{Ae}^2 / V_{ex}^2) * (c^2 / \omega_{pe}^2) * (1 / \delta_e^2)$$

For resuming

$$E_d / E_5 \sim (V_{Ae}^2 / V_{ex3}^2) * (\delta_e^2 / \delta_0^2)$$

The first factor is order one or larger, as discussed by the Authors.

The second factor is small only in "large-scale systems" for which $\delta_0 \gg \delta_e$, but it is of order one for "small-scale systems" with $\delta_0 \sim \delta_e$.

My conclusion is that the d) term must be retained, leading to a different scaling for E_{x1} , thus for V_{ex3} (outflow velocity) and, eventually, for the reconnection rate R , in particular for "small systems" (for which the Authors' predictions fails, as shown in Fig. 4).

Other points:

Lines 32-39 :

F. Califano et al., Front. Phys. 8, 317 (2020) is one of the first papers showing that the transition from standard reconnection to e-only reconnection occurs is related to the typical scale of current sheets (in that case, their size is related to the injection scale in magnetized turbulence). Citing this work would be fair.

Fig. 1b and Fig. 2 captions, as well as line 126 :

Please, carefully define the location at which the flaring angle "levels down". It is not, if I'm correct, the flex point of the separatrix, nor the point of the maximum value of the flaring angle. I agree that the definition of L_0 is arbitrary, but it should be clear.

Line 109 :

Is V_{Ae} defined at $x=0, z = d_e$? I guess it is at the time at which the reconnection rate is \sim cst and maximal, and not at $t=0$. Please, specify.

Line 182 :

The presence of E_x induces a variation of the pressure of the (nearly static) ions along the outflow. Ok.

Why should the electron pressure behave as the ion pressure ?

This point must be clarified. In particular, the Authors must corroborate the ion and electron pressure dynamics with the data of their simulations.

Line 192 : Please explain why $B_{z1} \sim (\delta_0/L_0)(B_{x6} / 2)$

Lines 256-258 :

Frankly speaking, the difference between the two curves is small.

Given the fact that both $f=0$ and $f=0.6$ cases are just estimations, and that the retroaction terms ($f \neq 0$) have been introduced ad hoc, I would not stress too much on the curve difference. I wonder if the f -terms add a benefit to this work.

Lines 307-309 :

Looking at Fig. 4g, it seems to me that theory predicts an R of ~ 0.5 , while simulations (blue stars) provide $R \sim 0.1$. Over-predictions are thus of around 0.4.

Minor points/Typos:

Fig. 1 caption:

Something lacks at line 3.

Fig. 1d :

Only 3 dots are present (final times ?, and not for $L_x = 1.28 d_i$), while the caption of Fig. 2 seems to suggest that dots in Fig. 1d are present at four different times, for each simulation curve.

Eq. (6) :

Should the last term be divided by (n^*e) ?

Reviewer #2

(Remarks to the Author)

See attached document

Reviewer #3

(Remarks to the Author)

Please, see my report attached.

Version 1:

Reviewer comments:

Reviewer #1

(Remarks to the Author)

The Authors have answered to all my point, and improved the readability of the manuscript.
The manuscript is now suitable for publication.

Reviewer #2

(Remarks to the Author)

See attachment

Reviewer #3

(Remarks to the Author)

Dear Authors, I thank you for your replies.

You have exhaustively addressed all my questions.

Also point (6) of my previous report, about which I had some doubts related to your approximation in present Eq. (4), is solved, and now I understand your statement.

I must apologize for what I know recognize to be just a partial understanding of your former Eq. (2) (now Eq. (3)), whose writing (and derivation strategy) I had confounded with that of former Eq. (3) (now Eq.(4)).

In general, the text is now clearer -although a little "dense" of information, I must admit. However, by considering the difficulty to communicate all the relevant details in the limited space available, I also compliment with you for the way the manuscript has been carefully constructed: all information can be ultimately found therein.

I can now recommend it for publication.

The only comment I have is that, if some space is left for writing, some further "explicit" information to the reader would be useful, about the way the finite difference approximation of present Eq. (3) is constructed: since you speak of "finite difference approximation", indeed, one could for example wonder if a sort of "2D compact difference stencil" was applied, and then about how it was formulated the way you do.

Personally I found helpful to look at Fig. 1c and Eq. (1) of your former work [34] (Liu et al., PRL (2017)), but I note that I did it once I went to look for hints in former papers of yours, since I still did not grasp on my own the approximation you used at r.h.s. of Eq. (3).

Just pointing explicitly to this former figure (or to other) of yours could be useful, I think -and even better would it be, if you manage to add some detailed info in the text, so to make the presentation more self-contained.

In any case, I do not need to see the manuscript again: I deem that any minor amendment in this sense -if any- could be dealt with a direct correspondence with the Editors.

We thank the reviewers for their valuable suggestions! We took this opportunity to significantly enhance the presentation, simulations and part of the theory. We are proud to present you with the revised version, and the modified theory now works much better. Especially, it now explains the upper bound value of the electron-only reconnection rate seen in simulations (see the updated Fig. 5), which motivated this work. The details of how we address your comments can be found below. The corresponding changes in the manuscript are highlighted in red color, and we provide the line numbers for the reviewers to find them easily.

Reviewer #1:

The manuscript is well written and the topic very interesting.

It has two weak points that, in my opinion, do not allow the publication on a broad-public journal as Physics Communications:

> We respectively hope that the reviewer can reconsider their position toward this work. We believe that we have addressed your concern, as detailed below. We are also confident that the current form of the theory is comprehensive.

1) Even if the subject is "broad public", since how fast magnetic reconnection proceed determines the dynamical temporal scale of a variety of magnetized system, the way the analytical estimations for the reconnection rate are presented is too specialistic. It is very hard for a reader, I would not say "not used to magnetic reconnection", but simply "not aware of the approach adopted in Ref. 28, 31-33" (cited by the Authors for justifying their approach) to understand the choice of different indexes for V_{ex} , B_x , etc.

I understand that the number of line is limited for a brief communication, but clearly state the geometry, method, approximations that lead to equation (2), as it has been done in the original articles using a similar technique, is mandatory. Otherwise, the article should be published on far more specialized journals, as JGR or PoP. Part of the Discussion, in particular the lines regarding Fig. 4e, 4d (that provide simple estimations, without numerical comparisons), as well as these two panels, could be sacrificed, at the advantage of a clear and pedagogical explanation of how getting Eq. (2).

> We thank you for the suggestions~ We have made the manuscript more friendly by carefully defining all the terms and providing clear physics intuition to the general readers. They are highlighted in various places on the first two pages.

In the previous version, we pointed out Refs. 28, 31-33, just to cite the success of this approach in other regimes. It is fine without mentioning these references for one to follow the theory.

This approach is simply the finite-difference approximation of the simplest form of the electron momentum equation (Eq. (1)) at one location. A reader who understands the basics of the electron momentum equation, Maxwell equations, and vector calculus should be able to follow the steps, and we have improved the accessibility of the derivation to non-experts in the revised version. This approach allows one to derive the algebraic relation between key quantities while taking into account the magnetic geometry.

We now explain these in lines 148-154. We also expand the discussion that leads to the original Eq. (2) (Now Eq. (3)) in lines 126-147, and also provide more intermediate steps to other equations throughout the paper. In the end, one would realize that it is a pretty simple, self-

contained theoretical model that captures various key features of this challenging, fully nonlinear problem.

We thank the reviewer's suggestion to eliminate the original Fig. 4e & 4d, which does free up space for pedagogical explanation, like the new figure 4!

2) I disagree with (part) of the estimation leading to Eq. (4).

The d) term in Eq. (3) is comparable with the main terms. I agree on the fact that far from the EDR its contribution is small (and roughly cancels the a) term. On the contrary, its contribution close to/in the EDR is far from negligible.

Indeed, even if B_x is small there, V_{ey} is large there. Their product roughly reads

$-J_y/(n * e) * B_x \sim - (B_{xe} * c)/(4\pi * \delta_e) * 1/(n * e) * B_{xe}$ where, as in the manuscript δ_e is the EDR width and B_{xe} is the value of B_x at the EDR scale.

Its integral contribution to Eq. (3) reads $-\int_0^{\delta_e} V_{ey} B_x dz \sim \int_0^{\delta_e} c * B_{xe}^2 / (4\pi * n * e * \delta_e) dz \sim c * B_{xe}^2 / (4\pi * n * e) \sim c * V_{Ae}^2 * (m_e / e)$

neglecting a factor of order one, coming from the actual profile of J_y and B_x .

Here $\int_0^{\delta_e} dz \sim \int_0^{\delta_e} dz$ since $V_{ey} B_x$ is small on the rest of the path, and V_{Ae} is, as in the manuscript, the value of V_{Aex} at the EDR scale.

Let's call E_d the contribution of the d) term to E_{x1} . It reads $E_d \sim c * V_{Ae}^2 * (m_e / e) * 2 / (c * L_0)$ Hereafter, I'll call E_5 the expression for E_{x1} provided by Eq (5). The ratio between E_d and E_5 reads

$E_d / E_5 \sim ((V_{Ae}^2 * m_e) / (L_0 * e)) / ((4\pi * n * e * \delta_0^2 * V_{ex3}^2) / (c^2 * L_0)) \sim (V_{Ae}^2 / V_{ex3}^2) * (c^2 / \omega_{pe}^2) * (1 / \delta_0^2)$

For resumming

$E_d / E_5 \sim (V_{Ae}^2 / V_{ex3}^2) * (\delta_e^2 / \delta_0^2)$

The first factor is order one or larger, as discussed by the Authors.

The second factor is small only in "large-scale systems" for which $\delta_0 \gg \delta_e$, but it is of order one for "small-scale systems" with $\delta_0 \sim \delta_e$.

My conclusion is that the d) term must be retained, leading to a different scaling for E_{x1} , thus for V_{ex3} (outflow velocity) and, eventually, for the reconnection rate R , in particular for "small systems" (for which the Authors' predictions fails, as shown in Fig. 4).

> Thanks for looking into the details. We understand the reviewer's argument, and we take this chance to consolidate our rationale. The key to eliminating this term is to consider the cancellation with another term. Terms (a) and (d) in Eq. (3) (Now Eq. (4)) roughly cancel each other because

$\int V_{ey} B_x dz \propto \int J_y B_x dz \propto \int (\partial_z B_x) B_x dz = \Delta(B_x^2)/2$ regardless of the detailed B_x profile, and the difference of B_x^2 between 4 and 3 (term (a)) is the same as that between 5 and 2 (negative term (b)) in our selection of the integral path. Thus, the elimination of both the (d) and (a) terms is reasonable. We made the change in lines 167-170.

Other points:

Lines 32-39 :

F. Califano et al., Front. Phys. 8, 317 (2020) is one of the first papers showing that the transition from standard reconnection to e-only reconnection occurs is related to the typical scale of

current sheets (in that case, their size is related to the injection scale in magnetized turbulence). Citing this work would be fair.

> Thanks for the suggestion. It is now cited in line 22 in the Introduction.

Fig. 1b and Fig. 2 captions, as well as line 126 :

Please, carefully define the location at which the flaring angle "levels down". It is not, if I'm correct, the flex point of the separatrix, nor the point of the maximum value of the flaring angle. I agree that the definition of L_0 is arbitrary, but it should be clear.

> We now clarify that the corners of the green boxes mark the locations downstream of which the exhaust opening angle quickly decreases to 0 in captions of Figs. 1 and 2, and line 123.

Line 109 :

Is V_{Ae} defined at $x=0, z = d_e$? I guess it is at the time at which the reconnection rate is $\sim cst$ and maximal, and not at $t=0$. Please, specify.

> V_{Ae} is basically the electron outflow speed at $x = L_e, z = 0$ as indicated in Fig. 1(b). Yes, it is defined at the nonlinear stage, where the reconnection rate reaches the plateau. We state this fact in line 102.

Line 182 :

The presence of E_x induces a variation of the pressure of the (nearly static) ions along the outflow. Ok. Why should the electron pressure behave as the ion pressure ?

This point must be clarified. In particular, the Authors must corroborate the ion and electron pressure dynamics with the data of their simulations.

> We realize that our discussion on electron pressure is confusing and not necessary; thus, we eliminate this sentence, especially since the electron pressure in our simulation is 10 times smaller than the ion pressure.

We now add a new panel as Fig. 3(b) to make our point. In this MMS motivated event, the ion temperature is very high compared to the in-plane Alfvénic energy, $T_{i0} = 115.16 m_i V_{Ai0}^2$ (as documented in the Methods section); thus, we are not able to fully suppress the oscillation caused by the PIC noise in the ΔP_{ixx} curve in green, even though we used both the time average and the Gaussian filter. Nevertheless, the signature of the pressure depletion and the pressure gradient are discernible. This is mentioned in the caption of Fig. 3(b) and lines 199-201.

After all, it is perceivable that the build-up of the back pressure will be significant if the reconnection X-line continues to pump plasmas into the O-line, especially within a small, closed system.

Line 192 : Please explain why $B_{z1} \sim (\delta_0/L_0)(B_{x6} / 2)$

> We apologize for the confusion. We now clarify it by introducing additional point "7" in Fig.3(a), which was omitted before. The original expression comes from $B_{z1} \simeq B_{z7} \simeq (\delta_0/L_0)B_{x7}$ and $B_{x7} \simeq B_{x6}/2$. We made the change in lines 218-219.

Lines 256-258 :

Frankly speaking, the difference between the two curves is small.

Given the fact that both $f=0$ and $f=0.6$ cases are just estimations, and that the retroaction terms ($f \neq 0$) have been introduced ad hoc, I would not stress too much on the curve difference. I wonder if the f -terms add a benefit to this work.

> Thanks for the suggestion. We fixed the way of how we include the back-pressure effect. We also imposed a reasonable condition where the outflow is completely shut off by the back-pressure in the $L_x \rightarrow 0$ limit, but not in the opposite limit. The resulting outflow (Eq. (8)) now works in various limits. Please check the discussion in lines 202-244. The black curves in Fig. 5 can explain the scaling of these key quantities now, importantly, the upper bound value $R \sim 1$ seen in simulations, which motivates this work.

Lines 307-309 :

Looking at Fig. 4g, it seems to me that theory predicts an R of ~ 0.5 , while simulations (blue stars) provide $R \sim 0.1$. Over-predictions are thus of around 0.4.

> We also note that the B_x at the location where $V_{ex,peak} = V_{Ae}$ holds accurately is roughly twice smaller than B_{xe} in the sharp B_x profile (that is proportional to the $B_x / \sqrt{4\pi n_e m_e}$ profile in Fig.1(d) because of the constant n_e). If we take this B_x as B_{xe} , the factor-of-two differences results in a four-times higher $R_{EDR} \propto (B_{xe}/B_{x0})^{-2}$, which may explain this discrepancy. Despite this extra complexity, our simple theory captures the constancy of the simulated R_{EDR} .

After all, the rate R is more important and there is no ambiguity in the choice of normalization. The rate R will be insensitive to the details of the EDR region as long as the localization mechanism is modeled. It is possible that introducing extra empirically tuned parameters can improve this prediction, but we like to keep the theory simple.

The improved discussion can be found in lines 334-347.

Minor points/Typos:

Fig. 1 caption:

Something lacks at line 3.

> We added “the” in line 3 of the caption. We also removed the “Slope” symbol in Fig. 1(a).

Fig. 1d :

Only 3 dots are present (final times ?, and not for $L_x = 1.28 d_i$), while the caption of Fig. 2 seems to suggest that dots in Fig. 1d are present at four different times, for each simulation curve.

> The pink dot is shown on the $L_x = 1.28 d_i$ case at the peak value.

Eq. (6) :

Should the last term be divided by (n^*e) ?

> Thanks for pointing out this typo. We now added the missing n^*e in the denominator of Eq.(6) (Now Eq.(7)).

Reviewer #2:

This paper presents an analytical model of electron-only reconnection and compares it to results from PIC simulations. Since its discovery a few years ago, electron reconnection has been a hot topic in the community and, given the previous work of the authors on analytical models of regular reconnection the extension in this paper is a natural fit.

While I found this paper timely and interesting, I do have several concerns (see below) that should be addressed before I can recommend it for publication.

Line 61: The possible implications of the quite large guide field (8 times the asymptotic field) should be discussed. I understand that the initial conditions are based on those in Pyakurel et al. (2019) which in turn come from Phan et al. (2018), but even in the former paper the relevance of this choice is not discussed. Is this strength guide field necessary for the analysis in the paper? For electron-only reconnection to occur? If a guide field of 8 is typical of electron-only reconnection it would be useful to note, and if it is not a discussion of whether the choice matters seems necessary.

> We thank the reviewer for bringing this up~

Even with a strong guide field ($B_g = 8B_{x0}$) in our simulation, the ion gyro-radius is comparable to the ion inertial scale, $\rho_i = 1.23d_i$. Guan et al.(2023) looked into both the strong and weak guide field limits and concluded that the $V_i \ll V_e$ condition is met when the system size is smaller than the ion gyro-radius (ρ_i). Presumably, because with a high ion thermal speed (e.g., $10.73V_{Ai0}$ in our runs) and large gyro-radii, ions will be quickly gyrotated out of the region of a constant \mathbf{E} , avoiding the formation of coherent ion flows through direct acceleration over a longer time span.

Our analytical theory is built on this $V_i \ll V_e$ condition (i.e., ions do not carry currents as in the EMHD limit), and it explains the transition to the standard rate around $L_x \gtrsim 10d_i$, as shown by Pyakurel et al. (2019). The analytical approach (and thus the predictions) derived here also works for the anti-parallel case under this condition, not limited to the strong guide field reconnection.

We add this discussions in lines 352-370.

Figure 1 caption: The “2.5” in the first line should be “2.56”.

> Thanks for pointing out this typo. It is now fixed.

Figure 1b: I appreciate that the ion velocity is so small as to be essentially invisible, but I am not convinced it's necessary to have essentially an image of nothing. Does rescaling the color bar so that a signal appears actually show a useful signal? Also, a rogue “6” seems to have escaped from Figure 3. (If its inclusion is intentional, there needs to be a mention of it in the caption or text.)

> Thanks for the suggestion. We now make the color bar range of Fig. 1(b) from $-V_{Ai0}$ to V_{Ai0} , which is the typical ion Alfvénic speed expected in standard reconnection and thus this color bar provides a meaningful plot. As expected, the ion flow indeed does not form and we take this

opportunity to annotate key quantities for our theoretical model on top of it. Label “6” is intended to compare with Fig. 3(a), and now it is explicitly mentioned in the caption.

Figure 1d: Since the reconnection rate is a key piece of the rest of the paper, it would be useful to give a brief description as to how it is determined. Difference in psi between the maximum and minimum values along the midplane? Out-of-plane electric field at the X-point?

> We now added the following in the caption. “..The rates in our simulations are computed from $R = (\partial\Delta\psi/\partial t)/B_{x0}V_{Ai0}$ where $\Delta\psi$ is the magnetic flux difference between the X-line and the O-line.”

Line 109: It would helpful to give the value of B_{xe} so readers can evaluate the correspondence with the actual outflow speed.

> Thanks for the suggestion. We now include a new panel (d) in figure 1 and discussion in lines 99-106 to show this relation.

Line 131: How does ion immobility matter here? The electron momentum equation doesn't include any ion terms.

> Thanks for the suggestion. In lines 133-137, we now clarify that the $V_i \ll V_e$ condition (i.e., ions do not carry currents) is used to replace \mathbf{V}_e using \mathbf{J} in the Lorentz force $-e\mathbf{V}_e \times \mathbf{B}/c$ in the electron momentum equation.

Line 141: Positive E_x only applies on one side of the X-line, does it not? More generally isn't it a field in the same direction as the outflow?

> Thanks for pointing this out. We now change the discussion to “...thus, we expect E_x pointing in the same direction of outflows, which slows down the electron jet.” in lines 146-147.

Line 147: While context suggests that “it” refers to the electron momentum equation, at first reading it seemed to refer to the $\mathbf{J} \times \mathbf{B}$ force.

> Thanks for pointing this out. We now change the sentence to “...To quantify this phenomenon, we simply discretize Eq. (2) at point “1” in Fig.3. In the x-direction,...” in lines 148-154.

Line 150: “corroborated” is probably not the intended word here. “along path 2-3, as shown in Figure 3” might be better.

> We now change it to lines 155-158.

Caption to Figure 3: It should probably be “Stokes loop” or something similar, since the namesake is George Stokes.

> Thanks for pointing out this typo. We now use “integral path” to reduce the use of terminology in this paper.

Line 168: Can a better justification for $B_{y0} \approx B_{y3}$ be given other than “it looks like that in the picture”? Returning to a previous comment, this could be a place where the assumption of a very strong guide field makes a critical difference.

> We now add a new panel as Fig. 3(b) to address the reviewer’s this and next comments together. As seen in the blue curves of Fig. 3(b), the variation of the $B_y^2/8\pi$ curve in blue is relatively small compared to $B_{x0}^2/8\pi$, further justifying our $B_{y0} \approx B_{y3}$ assumption. This is also discussed in lines 155-158.

Lines 177-180: Would assuming ion immobility affect the depletion of the pressure? Strictly speaking, if the ions cannot move their density cannot change, which means the only way the pressure could change is through a temperature variation. (And that’s only if there is a way to change the velocities of the ions without changing their positions.) Perhaps one may be able to argue that ion immobility is a zeroth-order assumption that might be violated at higher orders, but there’s no hint of that type of argument here. A more direct related question might be: Do the simulations show any variation in the ion pressure?

> We have made a more accurate statement on the ion immobility in lines 193-195 to reflect the reviewer’s point.

In addition, we now add a new panel as Fig. 3(b) to make our point. In this MMS motivated event, the ion temperature is very high compared to the in-plane Alfvénic energy, $T_{i0} = 115.16m_i V_{Ai0}^2$ (as documented in the Methods section); thus we are not able to fully suppress the oscillation caused by the PIC noise in the ΔP_{ixx} curve in green, even though we used both the time average and the Gaussian filter. Nevertheless, the signature pressure depletion and the build-up of the pressure gradient are discernible. This is discussed in the caption and lines 199-201.

After all, it is perceivable that the build-up of the back pressure will be significant, if the reconnection X-line continues to inject plasmas into the O-line, especially within a small, closed system.

Equation 7: The functional form suggests that the quantity under the root can be negative. Is that possible and, if so, what are the implications? Presumably it cannot happen for real reconnection (or the model breaks down as one approaches that limit).

> Thanks for the suggestion. We have now fixed the way of how we treat the back-pressure effect; basically, we now impose a system-size dependent condition where the outflow is completely shut off by the back-pressure in the $L_x \rightarrow 0$ limit, corresponding to the case where the back pressure completely cancels the magnetic tension that drives the reconnection process. In our new Eq. (8) for the limiting speed, the quantity inside the square root does not become negative within the reasonable range. Please check the discussion in lines 202-244. The black curves in Fig. 5 can explain the scaling of these key quantities now, especially the upper bound value of $R \sim 1$.

Line 196: If $de \gg \delta 0$, then the dependence on $\delta 0$ vanishes. That may not be a realistic limit, but it is a mathematically valid one that should probably be mentioned.

> Thanks for the suggestion! A discussion is added in lines 225-227. The updated Fig. 5(c) also shows this maximum plausible value V_{Ae0} now.

Lines 218-220: Returning to a previous comment, the only evidence given that the outflow speed approaches that based on B_{xe} is, I believe, the offhand statement in lines 108-109. It would be much easier to accept this statement with more detailed support.

> Thanks for the suggestion. We now include a new panel (d) in figure 1 and discussion in lines 99-106 to show this relation.

Line 251: The dashed black curves are located in Figure 4, but that's not clear from this sentence. It should be mentioned here rather than in the next line.

> Thanks for the suggestion. The discussion for Fig. 4 (Now Fig. 5) is now improved in lines 291-312.

Figure 4: Why do the red and blue stars for $L_x/d_i = 2.56$ not coincide? The parameters given in this paper seem identical to those in Pyakurel et al. (2019).

> We read off the values documented in Pyakurel (2019) for the purple symbols since we do not have the data anymore. In addition, in this work, we redo the runs that do not have exactly the same parameters as those in Pyakurel (2019).

Figure 4: Why does the horizontal axis extend to $L_x/d_i = 10$? The plotted points occupy less than 40% of the available space (and provoke questions as to why additional simulations weren't done to fill out the unused space).

> Thanks for the suggestion! we now performed one additional simulation of $L_x \times L_z = 7.68d_i \times 7.68d_i$ to fill the gap.

Line 259: The largest three simulations seem to at least somewhat match the predicted curves, but the degree to which the curves capture the scaling is debatable. The authors are entitled to take the optimistic view, but it's not immediately clear that, for instance, the rates in panel g actually increase with L_x or that the scaling in panel d matches. However, this is just a comment and not a demand that the authors change the text.

> In the revised version, we spent some time to extend the simulation domain size along the inflow direction for all cases of $L_x > 2.56d_i$. This avoids potential boundary effects from a fixed $L_z = 2.56d_i$ in the previous version. The overall scaling remains the same with some improvements in panel (d) (now Fig. 5(e)). For panel (g) (now Fig. 5(f)), we can only tell that the values are roughly constant. Most importantly, the maximum plausible reconnection rate R in Fig. 5(a) is on the order of 1, not 0.1 nor 10, while the limiting speed is only slightly super-ion-Alfvénic in Fig. 5(c). These are what we want to address in this work, as discussed in the section of "*Character of electron-only reconnection*".

Line 312: Returning again to B_{xe} , as noted above it would be helpful to have some of this discussion much earlier in the paper.

> Thanks for the suggestion. We now include a new panel (d) in figure 1 and discussion in lines 99-106 to show this relation.

The following items are minor grammatical issues

Lines 22-23: Either “reconnections have” or “reconnection has” – probably the former, given the “have” in lines 27

> Thanks for pointing this out. We now fixed it in line 23.

Line 29: “reconnection is their”

> Thanks for pointing this out. We now fixed it in line 29.

Line 117: “A similar Vex plateaus”

> Thanks for pointing this out. We now fixed it in line 114.

Line 159: “Terms” is wrong

> Thanks for pointing this out. We now fixed it in line 167.

Line 319-320: “suggests a reconnection rate be even higher”

> Thanks for the suggestion. We changed it accordingly in line 350.

Reviewer #3:

Dear Editor,

the interesting manuscript by Liu et al. discusses a modelling of the electron outflow speed in regimes of electron-only reconnection that are relevant to magnetospheric observations. The results are supported by numerical simulations and a critical comparison with previous numerical results and space-craft observations is given. The results discussed in this manuscript are new and surely useful for the interpretation of both in situ observations and nonlinear simulations. Therefore, it is with pleasure that I recommend it for publication on Communications Physics.

Nevertheless, before suggesting its acceptance, I have some remarks and questions, which I invite the Authors to consider for possible amendments: at least, some more specifications are needed, which would improve the readability of the text. Then, I have some (minor) “objection” about some points of the text. In any case, even if my objections were correct, I do not think they would much affect the results discussed in the manuscript, whose general relevance and soundness I recognize.

Dear Authors,

Here below my comments. Two of them (1-2) are of quite general character but are somewhat logically connected one each other. The other ones (3-7) are more punctual and refer to specific points of the text.

1) My first remark, of very general character, is related to the reconnection mechanism that you consider. I acknowledge that, so far, the identification of the so-called “electron-only” reconnection regime is still matter of discussion. Nevertheless, I note that a number of authors have suggested —with different emphasis— that electron-only reconnection may be related to reconnecting instability of tearing mode-type (and some have pointed also out its relevance to linear instabilities in the electron-MHD regime, which you also mention at line 323 of your work). Examples are provided by:

[Califano, F. et al. “Electron-only reconnection in plasma turbulence”. *Frontiers in Physics*, 8, 317 (2020)].

[Mallet, A. “The onset of electron-only reconnection”. *Journal of Plasma Physics*, 86, 905860301 (2020)].

[Franci, L., et al. “Anisotropic electron heating in turbulence-driven magnetic reconnection in the near-sun solar wind.” *The Astrophysical Journal* 936, 27 (2022)]

[Hubbert, M., et al. “Electron-only reconnection as a transition phase from quiet magnetotail current sheets to traditional magnetotail reconnection.” *Journal of Geophysical Research: Space Physics* 127, e2021JA029584 (2022)].

[Betar et al., “Asymptotic scalings of fluid, incompressible “electron-only” reconnection instabilities: Electron-magnetohydrodynamics tearing modes” 30, 072111 (2023)]

[Tsareva, O. O., et al. “Fast tearing mode driven by demagnetized electrons.” *Geophysical Research Letters* 51, e2023GL106867 (2024)].

In your whole manuscript you never mention this possibility, but I think it should merit at least some comment. It is true that the argument you propose “does not require” you to delve into the details of this aspect, but I note that some of the works above also provide —or suggest— theoretical models for electron-only reconnection, whose agreement with the model that you

propose is a priori not evident (cf. my remark (2), below).

I also note that, based on an argument that I am going to develop at point (2) below, the models which interpret electron-only reconnection as related to linear instabilities might even be “different”, or at least “complementary” to the model that you propose. I think you do not need to discuss this aspect in your work, but at the same time I think that this fact gives one more reason for which it is worth of mentioning, at least, the tearing-based point of view.

> Thanks for pointing out these interesting papers. As discussed in our response to your comment 2) below, we now discussed the potential role of tearing instability in lines 384-390. And, we cited these suggested references in appropriate places throughout this revision now.

2) My second remark, still of general character, is about the definition of reconnection rate which you use (e.g., in Eq.(12)), and which, besides, characterizes also the title of your work. Before to formulate my comment, please let me here make — for a reason that I am going to detail below— a kind of “logical distinction” between the definitions you use throughout your work. In this regard I would like to distinguish the first equality of Eq.(12) — that is, $R = cER/(Bx_0 V_{Ai0})$, — from the further approximations provided in terms of the ratio of fluid velocities, in which you identify for example $R \approx V_{in,e}/VA_e$ (Eq.(8)) or $R \approx V_{out,e}/VA_{i0}$ Slope (Eq.(12)), and I would like to compare all of them with the formal and “universal” definition $R \equiv (1/\Phi)(d\Phi/dt)$, where Φ is the magnetic flux defined as usual.

The definition $R = cER/(Bx_0 V_{Ai0})$ —consistent, e.g., with the one discussed in [Cassak, P. A., et al. "A review of the 0.1 reconnection rate problem", Journal of Plasma Physics 83, 715830501 (2017)]— seems to me to be generally compliant with the formal definition $R \equiv (1/\Phi)(d\Phi/dt)$. Instead, less evident appears to me the coherence in the most general case of the estimates of R provided in terms of the ratio of velocities (e.g., Eqs. (8) or last equalities of Eq. (12)), with respect to the definition $R \equiv (1/\Phi)(d\Phi/dt)$.

Of course, I am well aware that the estimates in terms of the velocity ratio are probably the most used ones, in literature, and I do not deny their usefulness for practical measures and for the characterization of different reconnection regimes —this is to say that I am not criticizing, here, your results or the relevance of your study. Nevertheless, in my understanding estimates like in Eq.(8) imply some steadiness condition during the reconnection process, which might not be so general. I mean: let us for a moment consider the possibility that a linear instability (e.g., a tearing mode) drives the electron-only reconnection process, according to the suggestions in the references quoted at my comment (1) above. While both $R = cER/(Bx_0 V_{Ai0})$ and $R \equiv (1/\Phi)(d\Phi/dt)$ give a reconnection rate which coincides with the growth rate of a tearing mode, it does not seem to me evident how $R \approx V_{in,e}/VA_e$ or $R \approx V_{out,e}/VA_{i0}$ Slope could do the same (cf., e.g., also comments going in this sense, about different definitions of the reconnection rates, in [Betar, H. et al, "Microscopic scales of linear tearing modes: a tutorial on boundary layer theory for magnetic reconnection." Journal of Plasma Physics 88, 925880601 (2022)]).

Do you have any comment/objection about this? Do you have any argument which can justify the generality of the final estimates of Eq.(8) or (12) also for non-steady reconnection processes?

If yes, I would invite you to mention it in the manuscript.

Otherwise, I would suggest you to better point out, somewhere in the text, the possible

general role of the “steadiness” assumption in your model — I say this also in the light of both my comment (1) above, and of the fact that you mention twice the relevance of “plasmoids” (line 177 and 284), which, in my understanding, are likely related to the onset of tearing-type instabilities, and you also explicitly speak of magnetic islands (lines 295-296). I also note that in several point you point out the role of steadiness (e.g., line 130, 151, 255, 266), but, it seems to me, you do not do it in the sense I just mentioned above, but just to justify some local algebraic passages or simplifications in the equations.

> Thanks for expressing your view and bring our attention to the nice work by Betar 2022. In our work, the dimensionless rate is defined as $R = (\partial\Delta\psi/\partial t)/B_{x0}V_{Ai0}$ where $\Delta\psi$ is the magnetic flux difference between the X-line and the O-line. The normalization factor in the denominator is constant for a given planner current sheet of a well-defined upstream condition, independent of how much the available upstream magnetic flux ψ is. I believe this is the key difference from the quantity $(\partial\Delta\psi/\partial t)/\psi$ suggested by the reviewer.

Our rate definition measures how fast the X-line processes the incoming magnetic flux (or magnetic field lines) in the simplest geometry. This definition is supported by the fact that this normalized rate is independent of the system size L_z along the inflow direction, which determines the available upstream ψ , as seen in simple current sheet simulations. And, this definition is consistent with that used in the classical Sweet-Parker model, Petschek model and our previous works, which we like to compare with.

In a 2D system (i.e., $\partial_y = 0$), this definition is also equivalent to $R = cE_y/B_{x0}V_{Ai0}$ at the X-line since the electrostatic component vanishes (i.e., $\partial_y\phi = 0$, where ϕ is the electric potential), and does not contribute to E_y .

In the quasi-steady state (i.e., $\partial_t = 0$) of a 2D system, the out-of-plane E_y is uniform because of Ampère’s law (i.e., $\partial_x E_y - \partial_y E_x = \partial_x E_y = 0$ and $\partial_z E_y - \partial_y E_z = \partial_z E_y = 0$). Reconnection is inclined to evolve into such a steady state, and the steadiness is quickly established at the X-line then the diffusion region. Thus, $E_y = V_{in}B_{x,in}/c$ is used in both the Sweet Parker and Petschek’s models. This constancy also spreads into the larger exhaust region if the system is large enough.

While this assumption appears to work for $L_x > 1.28d_i$ cases, this assumption indeed misses the unavoidable build-up of pressure at the O-line over time in the nonlinear saturation, especially for a system size $L_x \leq 1.28d_i$. That is why we added the correction from back-pressure to explain the stopped growing trend of the rate in the extremely small system limit. In revised version, the improved theory (updated in Fig. 5) can explain the upper bound value $R \sim 1$.

The reviewer is right that our approach also does not model the tearing instability, which is arguably important for the reconnection onset and the linear stage of reconnection. But note that we are more interested in the rate plateau (\sim steady state) observed in Fig. 1(e). We now make discussion on the potential role played by tearing physics in the linear stage, in lines 384-390.

Here below some more punctual remarks:

3) Lines 58-59 and Figs.1 and 3. I have a question, maybe naive, about your numerical results. You initialize the magnetic reconnection with a double Harris current sheet (lines 58-59), and Figs. 1,3 show the formation of a single magnetic island (due to the periodic boundary conditions you speak of at lines 75-76). Since your initial magnetic equilibrium formally corresponds to two neutral lines, I would have expected, instead, the onset of a kind of “double tearing” configuration, in the sense, e.g., of [Pritchett, P. L., et al., "Linear analysis of the double-tearing mode." *The Physics of Fluids* 23, 1368 (1980)]. Instead, frame (a) of Fig. 1 shows a single island with a single X-point and with the separatrices more or less positioned along the two “neutral lines”. Can you explain me why? I guess there is something I misunderstood.

> We stop our simulation before the growing plasmoid (the primary O-line) starts to compress the other current sheet, and that is why the upstream magnetic field remains straight in Fig. 1(a). Thus, the forcing from between the double-tearing modes in Pritchett (1980) is not relevant. In the revised version, we spent time conducting new simulations, confirming that the reconnection rates do not change with a wider domain size along the inflow direction. Note that this double current sheet setup in the periodic boundary is also widely used and the result is consistent with that of a single current sheet bounded by conductive boundaries.

4) Fig. 1: which operational definition did you use, here, to compute numerically the reconnection rate (frame (d))? Please specify it in the caption (cf. also my comment (2), above).

> Thanks for the suggestion! We now added “..The rates in our simulations are computed from $R = (\partial\Delta\psi/\partial t)/B_{x0}V_{Ai0}$ where $\Delta\psi$ is the magnetic flux difference between the X-line and the O-line.” in the caption.

5) Lines 147-148 and Eq. (2): I would suggest you to expand the sentence “...to discretize it in the x-direction”. I had to re-read it more than once together with the text below Eq.(2), and then to look at Figure 3, before to understand it. Maybe it could be sufficient just to add something like to “...to discretize it along a closed integration path, as sketched in Fig.3” – or something similar.

> Thanks for pointing out discussions that are confusing to readers. We now clarify the description using “...To quantify this phenomenon, we simply discretize Eq. (2) at point “1” in Fig.3. In the x-direction,...” in lines 148-154.

We also improved the discussion in various places to greatly enhance the accessibility of this manuscript to general readers.

Also, it is not evident to me the choice of estimating the first and third right hand side terms of Eq.(2) in the points (6) and (1), respectively. I mean: why exactly in these points? I can figure that “point (6)” is where the gradient along z is more important but.. I do not get the choice for (1), related to Ex. In any case, some more information in the text about these estimates would greatly help the reader.

> We apologize for this confusion related to your comment above. The approach is basically to get the finite-difference approximation of the electron momentum equation (now Eq. (2)) at the point “1”, and we analyze this equation in the x-direction to relate various key quantities surrounding this region. This simple, powerful approach allows one to incorporate the magnetic geometry inherent to a problem easily.

6) Lines 160-162 and Eq. (3): "...as B_x and V_{ey} only weakly depend on x outside the tiny EDR". Are you sure about this? I am not convinced of this statement, especially as far as the integration interval (3-4) is concerned: as you noted (lines 131-132) $V_{ey} \approx J_y$ because of the immobile ion assumption, and the out-of plane current density is known to have a peak along the separatrices of a magnetic island. Therefore, it seems to me that the term (a) be much more important than (d).

Could you support your statement with some numerical estimates or with more stringent argument? Otherwise, I would suggest you to accordingly amend the estimate (4).

> We take this chance to consolidate our rationale. The key to eliminating this term is to consider the cancellation with another term. Terms (a) and (d) roughly cancel each other because $\int V_{ey} B_x dz \propto \int J_y B_x dz \propto \int (\partial_z B_x) B_x dz = \Delta(B_x^2)/2$ regardless of the detailed B_x profile, and the difference of B_x^2 between 4 and 3 (term (a)) is the same as that between 5 and 2 (negative term (b)) in our selection of the integral path. Thus, the elimination of both the (d) and (a) terms is reasonable. We made the change in lines 167-170.

7) Eq. (4): I apologize but I have not understood some details. Namely:

- Why a factor 2 is dividing the electric field, at l.h.s.?

> The LHS of Eq. (4) (Now Eq.(5)) used the fact that E_x increases monotonically from 0 at the X-line to point "3", and this brings up a factor of 2. This is now indicated in lines 172-173.

- Why is E_x evaluated at (1)? (cf. also my comment (3), above);

> It is evaluated at point 1 because we like to estimate V_{ex3} using the finite difference approximation. This attempt should be clearer now in the discussion from lines 148-154.

- What is the origin of the first term in parenthesis at r.h.s.? Is it an algebraic average Why is B_{y6} evaluated there? (cf. also my comment (3), above). Please add some more info in the text.

> Yes, it is an algebraic average. The intermediate steps newly provided between lines 171-186 should be able to guide a reader to recover Eq.(5) and then Eq.(6) step by step.

We thank the reviewers for their final suggestions! The details of how we address their comments can be found below. The corresponding changes in the manuscript are highlighted in red, and we provide the line numbers so that one can find them easily.

Reviewer #1:

The Authors have answered to all my point, and improved the readability of the manuscript. The manuscript is now suitable for publication.

> Thank you!

Reviewer #2:

The authors have completely addressed the issues raised in my first review. I note some additional small points below found on readings of the revised version, but all of them are quite minor. In my opinion, this work is novel (very little theoretical analysis of electron-only reconnection has been published to date), technically sound, and strongly argued. I recommend it for publication in Communications Physics.

- The clause “of faster reconnection rates” in the first sentence of the abstract is awkward. Given the importance of its placement, it might be better to reword the entire sentence.

> We thank the reviewer’s suggestion! We improved the writing of the first sentence in the abstract.

- Figure 1: What is the purpose of the circles in panel e? Are these the “red symbols” mentioned in line 338? (If so, it should be noted that not all of the circles are red in this version.) Regardless, some mention of them should appear in the caption.

> Thanks for the reminder. These transparent circles are now explained in the caption.

- Line 115: This should probably be “four” simulations

> Thanks~ we changed it to “four” in line 128 of the revised version.

- Line 187: Is “genuinely” really meant here or is the intent closer to “generally”?

> We decided that “genuinely” or “generally” is not necessary for this sentence. Thus, we deleted it.

- Line 195: Neglecting P_e while including P_i is reasonable in this event since $T_e \ll T_i$, but is that true generally?

> As discussed in new lines 375-385, electron-only reconnection requires hot ions and cold electrons to remain frozen in the magnetic flux. Thus, $T_e \ll T_i$ is the relevant limit in this work. This statement on the ion momentum equation in line 195 (now line 215) remains accurate. We now add lines 438-440 to state this.

- Line 244: The symbol \gg seems to have been transformed into something else.

> The symbol \uparrow is intended, and the wording in new line 263 is improved to be consistent with it.

- Line 338: See the comment about Figure 1 above regarding the mysterious “red symbols” referred to here.

> In new line 359, we make our discussion on red symbols clearer.

- Line 355: The smallest guide field Guan et al consider is $B_g = 1$, which is arguably “weak” at best.

> Thanks for pointing this out~ We made the discussion of Guan’s work more accurate in new lines 378-379.

Reviewer #3

Dear Authors, I thank you for your replies.

You have exhaustively addressed all my questions.

Also point (6) of my previous report, about which I had some doubts related to your approximation in present Eq. (4), is solved, and now I understand your statement.

I must apologize for what I know recognize to be just a partial understanding of your former Eq. (2) (now Eq. (3)), whose writing (and derivation strategy) I had confounded with that of former Eq. (3) (now Eq.(4)).

In general, the text is now clearer -although a little "dense" of information, I must admit. However, by considering the difficulty to communicate all the relevant details in the limited space available, I also compliment with you for the way the manuscript has been carefully constructed: all information can be ultimately found therein.

I can now recommend it for publication.

The only comment I have is that, if some space is left for writing, some further "explicit" information to the reader would be useful, about the way the finite difference approximation of present Eq. (3) is constructed: since you speak of "finite difference approximation", indeed, one could for example wonder if a sort of "2D compact difference stencil" was applied, and then about how it was formulated the way you do.

Personally I found helpful to look at Fig. 1c and Eq. (1) of your former work [34] (Liu et al., PRL (2017)), but I note that I did it once I went to look for hints in former papers of yours, since I still did not grasp on my own the approximation you used at r.h.s. of Eq. (3).

Just pointing explicitly to this former figure (or to other) of yours could be useful, I think -and even better would it be, if you manage to add some detailed info in the text, so to make the presentation more self-contained.

In any case, I do not need to see the manuscript again: I deem that any minor amendment in this sense -if any- could be dealt with a direct correspondence with the Editors.

> We thank the reviewer's suggestion. We now add a few sentences pointing to the figure of our previous publication in new lines 168-171, which provide a comparison and nice contrast of what is new in this work.

Review of “An Analytical Model of ‘Electron-Only’ Magnetic Reconnection Rates”

Authors: Yi-Hin Liu, Prayash Pyakurel, Xiaocan Li, Michael Hesse, Naoki Bessho, Kevin Genestreti, and Shiva B. Thapa

Summary

This paper presents an analytical model of electron-only reconnection and compares it to results from PIC simulations. Since its discovery a few years ago, electron reconnection has been a hot topic in the community and, given the previous work of the authors on analytical models of regular reconnection the extension in this paper is a natural fit.

While I found this paper timely and interesting, I do have several concerns (see below) that should be addressed before I can recommend it for publication.

Detailed Concerns

- Line 61: The possible implications of the quite large guide field (8 times the asymptotic field) should be discussed. I understand that the initial conditions are based on those in Pyakurel et al. (2019) which in turn come from Phan et al. (2018), but even in the former paper the relevance of this choice is not discussed. Is this strength guide field necessary for the analysis in the paper? For electron-only reconnection to occur? If a guide field of 8 is typical of electron-only reconnection it would be useful to note, and if it is not a discussion of whether the choice matters seems necessary.
- Figure 1 caption: The “2.5” in the first line should be “2.56”.
- Figure 1b: I appreciate that the ion velocity is so small as to be essentially invisible, but I am not convinced it’s necessary to have essentially an image of nothing. Does rescaling the color bar so that a signal appears actually show a useful signal? Also, a rogue “6” seems to have escaped from Figure 3. (If its inclusion is intentional, there needs to be a mention of it in the caption or text.)
- Figure 1d: Since the reconnection rate is a key piece of the rest of the paper, it would be useful to give a brief description as to how it is determined. Difference in psi between the maximum and minimum values along the midplane? Out-of-plane electric field at the X-point?
- Line 109: It would helpful to give the value of B_{xe} so readers can evaluate the correspondence with the actual outflow speed.
- Line 131: How does ion immobility matter here? The electron momentum equation doesn’t include any ion terms.
- Line 141: Positive E_x only applies on one side of the X-line, does it not? More generally isn’t it a field in the same direction as the outflow?
- Line 147: While context suggests that “it” refers to the electron momentum equation, at first reading it seemed to refer to the $\mathbf{J} \times \mathbf{B}$ force.
- Line 150: “corroborated” is probably not the intended word here. “along path 2-3, as shown in Figure 3” might be better.
- Caption to Figure 3: It should probably be “Stokes loop” or something similar, since the namesake is George Stokes.

- Line 168: Can a better justification for $B_{y0} \approx B_{y3}$ be given other than “it looks like that in the picture”? Returning to a previous comment, this could be a place where the assumption of a very strong guide field makes a critical difference.
- Lines 177-180: Would assuming ion immobility affect the depletion of the pressure? Strictly speaking, if the ions cannot move their density cannot change, which means the only way the pressure could change is through a temperature variation. (And that’s only if there is a way to change the velocities of the ions without changing their positions.) Perhaps one may be able to argue that ion immobility is a zeroth-order assumption that might be violated at higher orders, but there’s no hint of that type of argument here. A more direct related question might be: Do the simulations show any variation in the ion pressure?
- Equation 7: The functional form suggests that the quantity under the root can be negative. Is that possible and, if so, what are the implications? Presumably it cannot happen for real reconnection (or the model breaks down as one approaches that limit).
- Line 196: If $d_e \gg \delta_0$, then the dependence on δ_0 vanishes. That may not be a realistic limit, but it is a mathematically valid one that should probably be mentioned.
- Lines 218-220: Returning to a previous comment, the only evidence given that the outflow speed approaches that based on B_{xe} is, I believe, the offhand statement in lines 108-109. It would be much easier to accept this statement with more detailed support.
- Line 251: The dashed black curves are located in Figure 4, but that’s not clear from this sentence. It should be mentioned here rather than in the next line.
- Figure 4: Why do the red and blue stars for $L_x/d_i = 2.56$ not coincide? The parameters given in this paper seem identical to those in Pyakurel et al. (2019).
- Figure 4: Why does the horizontal axis extend to $L_x/d_i = 10$? The plotted points occupy less than 40% of the available space (and provoke questions as to why additional simulations weren’t done to fill out the unused space).
- Line 259: The largest three simulations seem to at least somewhat match the predicted curves, but the degree to which the curves capture the scaling is debatable. The authors are entitled to take the optimistic view, but it’s not immediately clear that, for instance, the rates in panel g actually increase with L_x or that the scaling in panel d matches. However, this is just a comment and not a demand that the authors change the text.
- Line 312: Returning again to B_{xe} , as noted above it would be helpful to have some of this discussion much earlier in the paper.
- The following items are minor grammatical issues
 - Lines 22-23: Either “reconnections have” or “reconnection has” – probably the former, given the “have” in lines 27
 - Line 29: “reconnection is their”
 - Line 117: “A similar V_{ex} plateaus”
 - Line 159: “Terms” is wrong
 - Line 319-320: “suggests a reconnection rate be even higher”

Review of “An Analytical Model of ‘Electron-Only’ Magnetic Reconnection Rates”

Authors: Yi-Hin Liu, Prayash Pyakurel, Xiaocan Li, Michael Hesse, Naoki Bessho, Kevin Genestreti, and Shiva B. Thapa

Summary

The authors have completely addressed the issues raised in my first review. I note some additional small points below found on readings of the revised version, but all of them are quite minor. In my opinion, this work is novel (very little theoretical analysis of electron-only reconnection has been published to date), technically sound, and strongly argued. I recommend it for publication in *Communications Physics*.

Detailed Concerns

- The clause “of faster reconnection rates” in the first sentence of the abstract is awkward. Given the importance of its placement, it might be better to reword the entire sentence.
- Figure 1: What is the purpose of the circles in panel e? Are these the “red symbols” mentioned in line 338? (If so, it should be noted that not all of the circles are red in this version.) Regardless, some mention of them should appear in the caption.
- Line 115: This should probably be “four” simulations
- Line 187: Is “genuinely” really meant here or is the intent closer to “generally”?
- Line 195: Neglecting P_e while including P_i is reasonable in this event since $T_e \ll T_i$, but is that true generally?
- Line 244: The symbol \gg seems to have been transformed into something else.
- Line 338: See the comment about Figure 1 above regarding the mysterious “red symbols” referred to here.
- Line 355: The smallest guide field Guan et al consider is $B_g = 1$, which is arguably “weak” at best.

Report on the manuscript “An Analytical Model of “Electron-only” Magnetic Reconnection Rates”

Dear Editor,

the interesting manuscript by Liu et al. discusses a modelling of the electron outflow speed in regimes of electron-only reconnection that are relevant to magnetospheric observations. The results are supported by numerical simulations and a critical comparison with previous numerical results and space-craft observations is given. The results discussed in this manuscript are new and surely useful for the interpretation of both in situ observations and nonlinear simulations. Therefore, it is with pleasure that I recommend it for publication on Communications Physics.

Nevertheless, before suggesting its acceptance, I have some remarks and questions, which I invite the Authors to consider for possible amendments: at least, some more specifications are needed, which would improve the readability of the text. Then, I have some (minor) “objection” about some points of the text. In any case, even if my objections were correct, I do not think they would much affect the results discussed in the manuscript, whose general relevance and soundness I recognize.

Dear Authors,

Here below my comments. Two of them (1-2) are of quite general character but are somewhat logically connected one each other. The other ones (3-7) are more punctual and refer to specific points of the text.

1) My first remark, of very general character, is related to the reconnection mechanism that you consider. I acknowledge that, so far, the identification of the so-called “electron-only” reconnection regime is still matter of discussion. Nevertheless, I note that a number of authors have suggested —with different emphasis— that electron-only reconnection may be related to reconnecting instability of tearing mode-type (and some have pointed also out its relevance to linear instabilities in the electron-MHD regime, which you also mention at line 323 of your work). Examples are provided by:

- [Califano, F. et al. “*Electron-only reconnection in plasma turbulence*”. *Frontiers in Physics*, **8**, 317 (2020)].
- [Mallet, A. “*The onset of electron-only reconnection*”. *Journal of Plasma Physics*, **86**, 905860301 (2020)].
- [Franci, L., et al. “*Anisotropic electron heating in turbulence-driven magnetic reconnection in the near-sun solar wind*.” *The Astrophysical Journal* **936**, 27 (2022)]
- [Hubbert, M., et al. “*Electron-only reconnection as a transition phase from quiet magnetotail current sheets to traditional magnetotail reconnection*.” *Journal of Geophysical Research: Space Physics* **127**, e2021JA029584 (2022)].
- [Betar et al., “*Asymptotic scalings of fluid, incompressible “electron-only” reconnection instabilities: Electron-magnetohydrodynamics tearing modes*” **30**, 072111 (2023)]
- [Tsareva, O. O., et al. “*Fast tearing mode driven by demagnetized electrons*.” *Geophysical Research Letters* **51**, e2023GL106867 (2024)].

In your whole manuscript you never mention this possibility, but I think it should merit at least some comment. It is true that the argument you propose “does not require” you to delve into the details of this aspect, but I note that some of the works above also provide —or suggest— theoretical models for electron-only reconnection, whose agreement with the model that you propose is a priori not evident (cf. my remark (2), below).

I also note that, based on an argument that I am going to develop at point (2) below, the models which interpret electron-only reconnection as related to linear instabilities might even be “different”, or at least “complementary” to the model that you propose. I think you do not need to discuss this aspect in your work, but at the same time I think that this fact gives one more reason for which it is worth of mentioning, at least, the tearing-based point of view.

2) My second remark, still of general character, is about the definition of reconnection rate which you use (e.g., in Eq.(12)), and which, besides, characterizes also the title of your work.

Before to formulate my comment, please let me here make — for a reason that I am going to detail below — a kind of “logical distinction” between the definitions you use throughout your work. In this regard I would like to distinguish the first equality of Eq.(12) — that is, $R = cE_R / (B_{x0} V_{ai0})$, — from the further approximations provided in terms of the ratio of fluid velocities, in which you identify for example $R \approx V_{in,e} / V_{Ae}$ (Eq.(8)) or $R \approx V_{out,e} / V_{Ai0} S_{lope}$ (Eq.(12)), and I would like to compare all of them with the formal and “universal” definition $R = (1/\Phi)(d\Phi/dt)$, where Φ is the magnetic flux defined as usual.

The definition $R = cE_R / (B_{x0} V_{ai0})$ — consistent, e.g., with the one discussed in [Cassak, P. A., et al. "A review of the 0.1 reconnection rate problem", *Journal of Plasma Physics* **83**, 715830501 (2017)] — seems to me to be generally compliant with the formal definition $R = (1/\Phi)(d\Phi/dt)$. Instead, less evident appears to me the coherence in the most general case of the estimates of R provided in terms of the ratio of velocities (e.g., Eqs. (8) or last equalities of Eq. (12)), with respect to the definition $R = (1/\Phi)(d\Phi/dt)$.

Of course, I am well aware that the estimates in terms of the velocity ratio are probably the most used ones, in literature, and I do not deny their usefulness for practical measures and for the characterization of different reconnection regimes — this is to say that I am not criticizing, here, your results or the relevance of your study. Nevertheless, in my understanding estimates like in Eq.(8) imply some steadiness condition during the reconnection process, which *might not* be so general. I mean: let us for a moment consider the possibility that a linear instability (e.g., a tearing mode) drives the electron-only reconnection process, according to the suggestions in the references quoted at my comment (1) above. While both $R = cE_R / (B_{x0} V_{ai0})$ and $R = (1/\Phi)(d\Phi/dt)$ give a reconnection rate which coincides with the growth rate of a tearing mode, it does not seem to me evident how $R \approx V_{in,e} / V_{Ae}$ or $R \approx V_{out,e} / V_{Ai0} S_{lope}$ could do the same (cf., e.g., also comments going in this sense, about different definitions of the reconnection rates, in [Betar, H. et al, "Microscopic scales of linear tearing modes: a tutorial on boundary layer theory for magnetic reconnection." *Journal of Plasma Physics* **88**, 925880601 (2022)]).

Do you have any comment/objection about this? Do you have any argument which can justify the generality of the final estimates of Eq.(8) or (12) also for non-steady reconnection processes?

If yes, I would invite you to mention it in the manuscript.

Otherwise, I would suggest you to better point out, somewhere in the text, the possible general role of the “steadiness” assumption in your model — I say this also in the light of both my comment (1) above, and of the fact that you mention twice the relevance of “plasmoids” (line 177 and 284), which, in my understanding, are likely related to the onset of tearing-type instabilities, and you also explicitly speak of magnetic islands (lines 295-296). I also note that in several point you point out the role of steadiness (e.g., line 130, 151, 255, 266), but, it seems to me, you do not do it in the sense I just mentioned above, but just to justify some local algebraic passages or simplifications in the equations.

Here below some more punctual remarks:

3) **Lines 58-59 and Figs.1 and 3.** I have a question, maybe naive, about your numerical results. You initialize the magnetic reconnection with a double Harris current sheet (lines 58-59), and Figs. 1,3 show the formation of a single magnetic island (due to the periodic boundary conditions you speak of at lines 75-76). Since your initial magnetic equilibrium formally corresponds to two neutral lines, I would have expected, instead, the onset of a kind of “double tearing” configuration, in the sense, e.g., of [Pritchett, P. L., et al., "Linear analysis of the double-tearing mode." *The Physics of Fluids* **23**, 1368 (1980)]. Instead, frame (a) of Fig. 1 shows a single island with a single X-point and with the separatrices more or less positioned along the two “neutral lines”. Can you explain me why? I guess there is something I misunderstood.

4) **Fig. 1:** which operational definition did you use, here, to compute numerically the reconnection rate (frame (d))? Please specify it in the caption (cf. also my comment (2), above).

- 5) **Lines 147-148 and Eq. (2):** I would suggest you to expand the sentence “...to discretize it in the x -direction”. I had to re-read it more than once together with the text below Eq.(2), and then to look at Figure 3, before to understand it. Maybe it could be sufficient just to add something like to “...to discretize it along a closed integration path, as sketched in Fig.3” — or something similar.

Also, it is not evident to me the choice of estimating the first and third right hand side terms of Eq.(2) in the points (6) and (1), respectively. I mean: why exactly in these points? I can figure that “point (6)” is where the gradient along z is more important but.. I do not get the choice for (1), related to E_x . In any case, some more information in the text about these estimates would greatly help the reader.

- 6) **Lines 160-162 and Eq. (3):** “...as B_x and V_{ey} only weakly depend on x outside the tiny EDR”. Are you sure about this? I am not convinced of this statement, especially as far as the integration interval (3-4) is concerned: as you noted (lines 131-132) $V_{ey} \approx J_y$ because of the immobile ion assumption, and the out-of plane current density is known to have a peak along the separatrices of a magnetic island. Therefore, it seems to me that the term (a) be much more important than (d).

Could you support your statement with some numerical estimates or with more stringent argument? Otherwise, I would suggest you to accordingly amend the estimate (4).

- 7) **Eq. (4):** I apologize but I have not understood some details. Namely:
- Why a factor 2 is dividing the electric field, at l.h.s.?
 - Why is E_x evaluated at (1)? (cf. also my comment (3), above);
 - What is the origin of the first term in parenthesis at r.h.s.? Is it an algebraic average? Why is B_{y6} evaluated there? (cf. also my comment (3), above).
- Please add some more info in the text.